# Learning Student-Friendly Teacher Networks for Knowledge Distillation

Dae Young Park[*,1], Moon-Hyun Cha[1], Changwook Jeong[1], Dae Sin Kim[1], and Bohyung Han[*,2]

[1]DIT Center, Samsung Electronics, Korea
[2]ECE & ASRI, Seoul National University, Korea
{p30.daeyoung, moonhyun.cha, chris.jeong, daesin.kim}@samsung.com
bhhan@snu.ac.kr

## Abstract

We propose a novel knowledge distillation approach to facilitate the transfer of dark knowledge from a teacher to a student. Contrary to most of the existing methods that rely on effective training of student models given pretrained teachers, we aim to learn the teacher models that are friendly to students and, consequently, more appropriate for knowledge transfer. In other words, at the time of optimizing a teacher model, the proposed algorithm learns the student branches jointly to obtain student-friendly representations. Since the main goal of our approach lies in training teacher models and the subsequent knowledge distillation procedure is straightforward, most of the existing knowledge distillation methods can adopt this technique to improve the performance of diverse student models in terms of accuracy and convergence speed. The proposed algorithm demonstrates outstanding accuracy in several well-known knowledge distillation techniques with various combinations of teacher and student models even in the case that their architectures are heterogeneous and there is no prior knowledge about student models at the time of training teacher networks.

## 1 Introduction

Knowledge distillation [1] is a well-known technique to learn compact deep neural network models with competitive accuracy, where a smaller network (student) is trained to simulate the representations of a larger one (teacher). The popularity of knowledge distillation is mainly due to its simplicity and generality; it is straightforward to learn a student model based on a teacher and there is no restriction about the network architectures of both models. The main goal of most approaches is how to transfer dark knowledge to student models effectively, given predefined and pretrained teacher networks.

Although knowledge distillation is a promising and convenient method, it sometimes fails to achieve satisfactory performance in terms of accuracy. This is partly because the model capacity of a student is too limited compared to that of a teacher and knowledge distillation algorithms are suboptimal [2, 3]. In addition to this reason, we claim that the consistency of teacher and student features is critical to knowledge transfer and the inappropriate representation learning of a teacher often leads to the suboptimality of knowledge distillation.

We are interested in making a teacher network hold better transferable knowledge by providing the teacher with a snapshot of the student model at the time of its training. We take advantage of the typical structures of convolutional neural networks with multiple blocks and make the representations of each block in teachers easy to be transferred to students. The proposed approach aims to train

---

[*]Equal contribution

35th Conference on Neural Information Processing Systems (NeurIPS 2021).

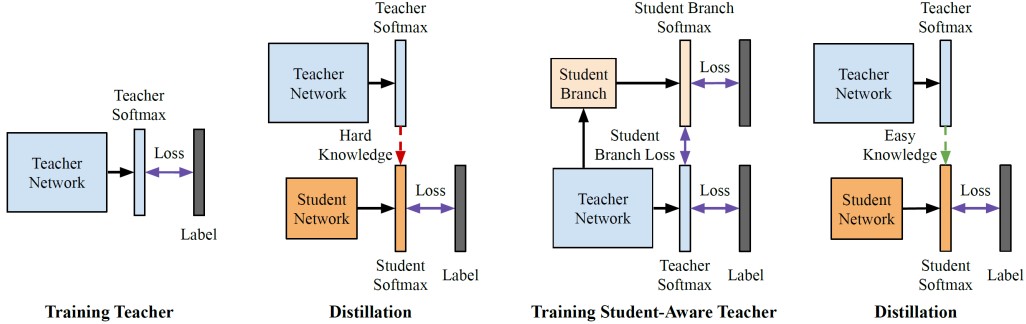

(a) Standard knowledge distillation    (b) Student-friendly teacher network

Figure 1: Comparison between the standard knowledge distillation and our approach. (a) The standard knowledge distillation trains teachers alone and then distill knowledge to students. (b) The proposed student-friendly teacher network trains teachers along with student branches, and then distill more easy-to-transfer knowledge to students.

teacher models friendly to students for facilitating knowledge distillation; we call the teacher model trained by this strategy student-friendly teacher network (SFTN). SFTN is deployed in arbitrary distillation algorithms easily due to its generality for training models and transferring knowledge.

SFTN is partly related to collaborative learning methods [4, 5, 6], which may suffer from the correlation between the models trained jointly and fail to fully exploit knowledge in teacher models. On the other hand, SFTN is free from the limitation since it performs knowledge transfer from a teacher to a student in one direction via a two-stage learning procedure—student-aware training of teacher network followed by knowledge distillation from a teacher to a student. Although the structure of a teacher network depends on target student models, it is sufficiently generic to be adopted by students with various architectures. Figure 1 demonstrates the main difference between the proposed algorithm and the standard knowledge distillation methods.

The following is the list of our main contributions:

- We adopt a student-aware teacher learning procedure before knowledge distillation, which enables teacher models to transfer their representations to students more effectively.

- The proposed approach is applicable to diverse architectures of teacher and students while it can be incorporated into various knowledge distillation algorithms.

- We demonstrate that the integration of SFTN into various baseline algorithms and models improve accuracy consistently with substantial margins.

The rest of the paper is organized as follows. We first discuss the existing knowledge distillation techniques in Section 2. Section 3 describes the details of the proposed SFTN including the knowledge distillation algorithm. The experimental results with in-depth analyses are presented in Section 4, and we make the conclusion in Section 5.

## 2 Related Work

Although deep learning has shown successful outcomes in various fields, it is still difficult to apply deep neural networks to real-world tasks due to their excessive requirement for computation and memory. There have been many attempts to reduce the computational cost of deep learning models, and knowledge distillation is one of the examples. Various computer vision [7, 8, 9, 10] and natural language processing [11, 12, 13, 14] tasks often employ knowledge distillation to obtain efficient models. Recently, some cross-modal tasks [15, 16, 17] transfer knowledge across domains. This section summarizes the research efforts to improve performance of models via knowledge distillation.

### 2.1 What to distill

Since Hinton et al. [1] introduce the basic concept of knowledge distillation, where the dark knowledge in teacher models is given by the temperature-scaled representations of the softmax function, various

kinds of information have been employed as the sources of knowledge for distillation from teachers to students. FitNets [18] distills intermediate features of a teacher network, where the student network transforms the intermediate features using guided layers and then calculates the difference between the guided layers and the intermediate features of teacher network. The position of distillation is shifted to the layers before the ReLU operations in [19], which also proposes the novel activation function and the partial $L_2$ loss function for effective knowledge transfer. Zagoruyko and Komodakis [20] argue importance of attention and propose an attention transfer (AT) method from teachers to students while Kim et al. [21] compute the factor information of the teacher representations using an autoencoder, which is decoded by students for knowledge transfer. Relational knowledge distillation (RKD) [22] introduces a technique to transfer relational information such as distances and angles of features.

CRD [23] maximizes mutual information between a teacher and a student via contrastive learning. There exist a couple of methods to perform knowledge distillation without teacher models. For example, ONE [24] distills knowledge from an ensemble of multiple students while BYOT [25] transfers knowledge from deeper layers to shallower ones. Besides, SSKD [26] distills self-supervised features of teachers to students for transferring richer knowledge.

## 2.2 How to distill

Several recent knowledge distillation methods focus on the strategy of knowledge distillation. Born again network (BAN) [27] presents the effectiveness of sequential knowledge distillation via the networks with an identical architecture. A curriculum learning method [28] employs the optimization trajectory of a teacher model to train students. Collaborative learning approaches [4, 5, 6] attempt to learn multiple models with distillation jointly, but their concept is not well-suited for asymmetric teacher-student relationship, which may lead to suboptimal convergence of student models.

The model capacity gap between a teacher and a student is addressed in [2, 29, 3]. TAKD [3] employs an extra network to reduce model capacity gap between teacher and student models, where a teacher transfers knowledge to a student via a teaching assistant network with an intermediate size. An early stopping technique for training teacher networks is proposed to obtain better transferable representations and a neural architecture search is employed to identify a student model with the optimal size [2]. Our work proposes a novel student-friendly learning technique of a teacher network to facilitate knowledge distillation.

# 3 Student-Friendly Knowledge Distillation

This section describes the details of the student-friendly teacher network (SFTN), which transfers the features of teacher models to student networks more effectively than the standard distillation. Figure 2 illustrates the main idea of our method.

## 3.1 Overview

The conventional knowledge distillation approaches attempt to find the way of teaching student networks given the architecture of teacher networks. The teacher network is trained with the loss with respect to the ground-truth, but the objective is not necessarily beneficial for knowledge distillation to students. To the contrary, SFTN framework aims to improve the effectiveness of knowledge distillation from the teacher to the student models.

**Modularizing teacher and student networks** We modularize teacher and student networks into multiple blocks based on the depth of layers and the feature map sizes. This is because knowledge distillation is often performed at every 3 or 4 blocks for accurate extraction and transfer of knowledge in teacher models. Figure 2 presents the case that both networks are modularized into 3 blocks, denoted by $\{B_T^1, B_T^2, B_T^3\}$ and $\{B_S^1, B_S^2, B_S^3\}$ for a teacher and a student, respectively.

**Adding student branches** SFTN augments student branches to a teacher model for the joint training of both parts. Each student branch is composed of a teacher network feature transform layer $\mathcal{T}$ and a student network blocks. Note that $\mathcal{T}$ is similar to a guided layer in FitNets [18] and transforms the dimensionality of the channel in $\mathbf{F}_T^i$ into that of $B_S^{i+1}$. Depending on the configuration of teacher and student networks, the transformation need to increase or decrease the size of the feature maps. We employ 3×3 convolutions to reduce the size of $\mathbf{F}_T^i$ while 4×4 transposed convolutions are

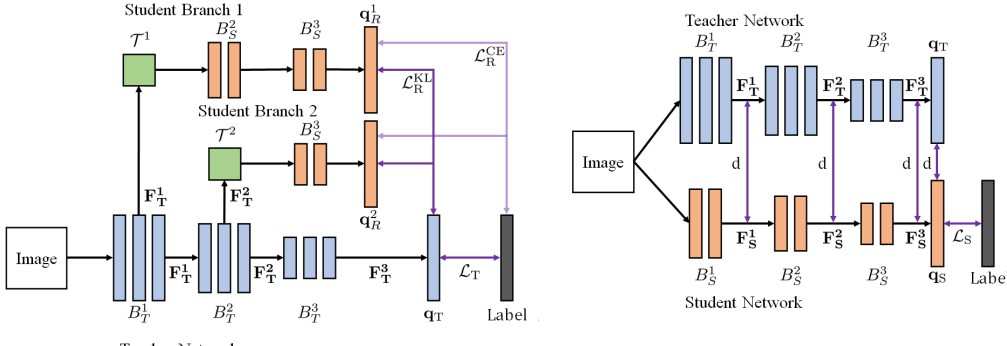

| (a) Student-aware training of a teacher network | (b) Knowledge distillation |

Figure 2: Overview of the student-friendly teacher network (SFTN). In this figure, $\mathbf{F}$, $B$, $\mathcal{T}$, and $\mathbf{q}$ denote a feature map, a network block, a teacher network feature transform layer, and a softmax output, respectively, where the superscript means the network block index and the subscript S, T, and R respectively indicate the student network, the teacher network, and the student branch in the teacher model. The loss for teacher network $\mathcal{L}_T$ is given by (4) while Kullback-Leibler loss $\mathcal{L}_R^{KL}$ and cross entropy loss $\mathcal{L}_R^{CE}$ are defined in (5) and (6), respectively. (a) When training a teacher, SFTN optimizes $\mathbf{F}_T^i$ and $\mathbf{q}_T$ for better knowledge transfer to student networks. (b) In the distillation stage, the features in the teacher network, $\mathbf{F}_T^i$ and $\mathbf{q}_T$, are distilled to student networks with existing knowledge distillation algorithms straightforwardly.

used to increase its size. Also, $1 \times 1$ convolutions is used when we do not need to change the size of $\mathbf{F}_T^i$. The features transformed to a student branch is forwarded separately to compute the logit of the branch. For example, as shown in Figure 2(a), $\mathbf{F}_T^1$ in the teacher stream is transformed to fit $B_S^2$, which initiates a student branch to derive $\mathbf{q}_R^1$ while another student branch starts from the transformed features of $\mathbf{F}_T^2$. Note that $\mathbf{F}_T^3$ has no trailing teacher network block in the figure and has no associated student branch because it is directly utilized to compute the logit of the main teacher network.

**Training SFTN**    The teacher network is trained along with multiple student branches corresponding to individual blocks in the teacher, where we minimize the differences in the representations between the teacher and the student branches. Our loss function is composed of three terms: loss in the teacher network $\mathcal{L}_T$, Kullback-Leibler loss $\mathcal{L}_R^{KL}$ in the student branch, and cross-entropy loss $\mathcal{L}_R^{CE}$ in the student branch. The main loss term, $\mathcal{L}_T$, minimizes the error between $\mathbf{q}_T$ and the ground-truth while $\mathcal{L}_R^{KL}$ enforces $\mathbf{q}_R^i$ and $\mathbf{q}_T$ to be similar to each other and $\mathcal{L}_R^{CE}$ makes $\mathbf{q}_R^i$ fit the ground-truth.

**Distillation using SFTN**    As shown in Figure 2(b), the conventional knowledge distillation technique is employed to simulate $\mathbf{F}_T^i$ and $\mathbf{q}_T$ by $\mathbf{F}_S^i$ and $\mathbf{q}_S$ respectively. The actual knowledge distillation step is straightforward because the representations of $\mathbf{F}_T^i$ and $\mathbf{q}_T$ have already been learned properly at the time of training SFTN. We expect the performance of the student network distilled from the SFTN to be better than the one obtained from the conventional teacher network.

### 3.2   Network Architecture

SFTN consists of a teacher network and multiple student branches. The teacher and student networks are divided into $N$ blocks, where a set of blocks in the teacher is given by $\mathbb{B}_T = \{B_T^i\}_{i=1}^N$ while the blocks in the student is denoted by $\mathbb{B}_S = \{B_S^i\}_{i=1}^N$. Note that the last block in the teacher network does not have the associated student branch.

Given an input of the network, $\mathbf{x}$, the output of the softmax function for the main teacher network, $\mathbf{q}_T$, is given by

$$\mathbf{q}_T(\mathbf{x}; \tau) = \mathrm{softmax}\left(\frac{\mathcal{F}_T(\mathbf{x})}{\tau}\right), \tag{1}$$

where $\mathcal{F}_T$ denotes the logit of the teacher network and $\tau$ is the temperature of the softmax function. On the other hand, the output of the softmax function in the $i^{\text{th}}$ student branch, $\mathbf{q}_R^i$, is given by

$$\mathbf{q}_R^i(\mathbf{F}_T^i; \tau) = \text{softmax}\left(\frac{\mathcal{F}_S^i(\mathcal{T}^i(\mathbf{F}_T^i))}{\tau}\right),\tag{2}$$

where $\mathcal{F}_S^i$ denotes the logit of the $i^{\text{th}}$ student branch.

### 3.3 Loss Functions

The teacher network in the conventional knowledge distillation framework is traned only with $\mathcal{L}_T$. However, SFTN has additional loss terms such as $\mathcal{L}_R^{\text{KL}}$ and $\mathcal{L}_R^{\text{CE}}$ as described in Section 3.1. The total loss function of SFTN, denoted by $\mathcal{L}_{\text{SFTN}}$, is given by

$$\mathcal{L}_{\text{SFTN}} = \lambda_T \mathcal{L}_T + \lambda_R^{\text{KL}} \mathcal{L}_R^{\text{KL}} + \lambda_R^{\text{CE}} \mathcal{L}_R^{\text{CE}},\tag{3}$$

where $\lambda_T$, $\lambda_R^{\text{KL}}$ and $\lambda_R^{\text{CE}}$ are the weights of individual loss terms.

Each loss term is defined as follows. First, $\mathcal{L}_T$ is given by the cross-entropy between the teacher's prediction $\mathbf{q}_T$ and the ground-truth label $\mathbf{y}$ as

$$\mathcal{L}_T = \text{CrossEntropy}(\mathbf{q}_T, \mathbf{y})\tag{4}$$

The knowledge distillation loss, denoted by $\mathcal{L}_R^{\text{KL}}$, employs the KL divergence between $\mathbf{q}_R^i$ and $\mathbf{q}_T$, where $N-1$ student branches except for the last block in the teacher network are considered together as

$$\mathcal{L}_R^{\text{KL}} = \frac{1}{N-1}\sum_{i=1}^{N-1} \text{KL}(\tilde{\mathbf{q}}_R^i \| \tilde{\mathbf{q}}_T),\tag{5}$$

where $\tilde{\mathbf{q}}_R^i$ and $\tilde{\mathbf{q}}_T$ denote smoother softmax function outputs with a larger temperature, $\tilde{\tau}$. The cross-entropy loss of the student network, $\mathcal{L}_R^{\text{CE}}$, is obtained by the average cross-entropy loss from all the student branches, which is given by

$$\mathcal{L}_R^{\text{CE}} = \frac{1}{N-1}\sum_{i=1}^{N-1} \text{CrossEntropy}(\mathbf{q}_R^i, \mathbf{y}).\tag{6}$$

Note that we set $\tau$ to 1 for both the cross-entropy loss, $\mathcal{L}_T$ and $\mathcal{L}_R^{\text{CE}}$.

## 4 Experiments

We evaluate the performance of SFTN in comparison to existing methods and analyze the characteristics of SFTN in various aspects. We first describe our experiment setting in Section 4.1. Then, we compare results between SFTNs and the standard teacher networks with respect to classification accuracy in various knowledge distillation algorithms in Section 4.2. The results from ablative experiments for SFTN and transfer learning are discussed in the rest of this section.

### 4.1 Experiment Setting

We perform evaluation on multiple well-known datasets including ImageNet [30] and CIFAR-100 [31] using several different backbone networks such as ResNet [32], WideResNet [33], VGG [34], ShuffleNetV1 [35], and ShuffleNetV2 [36]. For comprehensive evaluation, we adopt various knowledge distillation techniques, which include KD [1], FitNets [18], AT [20], SP [37], VID [38], RKD [22], PKT [39], AB [40], FT [21], CRD [23], SSKD [26], and OH [19]. Among these methods, the feature distillation methods [18, 20, 37, 38, 22, 39, 40, 21, 19] conduct joint distillation with conventional KD [1] during student training, which results in higher accuracy in practice than the feature distillation only. We also include comparisons with collaborative learning methods such as DML [4] and KDCL [5], and a curriculum learning technique, RCO [28]. We have reproduced the results from the existing methods using the implementations provided by the authors of the papers.

Table 1: Comparisons between SFTN and the standard teacher models on CIFAR-100 dataset when the architectures of the teacher-student pairs are homogeneous. In all the tested algorithms, the students distilled from the teacher models given by SFTN outperform the ones trained from the standard teacher models. All the reported results are based on the outputs of 3 independent runs.

| Teacher/Student Teacher training | WRN40-2/WRN16-2 | | | WRN40-2/WRN40-1 | | | resnet32x4/resnet8x4 | | | resnet32x4/resnet8x2 | | |
|---|---|---|---|---|---|---|---|---|---|---|---|---|
| | Stan. | SFTN | Δ | Stan. | SFTN | Δ | Stan. | SFTN | Δ | Stan. | SFTN | Δ |
| Teacher Acc. | 76.30 | 78.20 | | 76.30 | 77.62 | | 79.25 | 79.41 | | 79.25 | 77.89 | |
| Student Acc. w/o KD | | 73.41 | | | 72.16 | | | 72.38 | | | 68.19 | |
| KD [1] | 75.46 | 76.25 | +0.79 | 73.73 | 75.09 | +1.36 | 73.39 | 76.09 | +2.70 | 67.43 | 69.17 | +1.74 |
| FitNets [18] | 75.72 | 76.73 | +1.01 | 74.14 | 75.54 | +1.40 | 75.34 | 76.89 | +1.55 | 69.80 | 71.07 | +1.27 |
| AT [20] | 75.85 | 76.82 | +0.97 | 74.56 | 75.86 | +1.30 | 74.98 | 76.91 | +1.93 | 68.79 | 70.90 | +2.11 |
| SP [37] | 75.43 | 76.77 | +1.34 | 74.51 | 75.31 | +0.80 | 74.06 | 76.37 | +2.31 | 68.39 | 70.03 | +1.64 |
| VID [38] | 75.63 | 76.79 | +1.16 | 74.21 | 75.76 | +1.55 | 74.86 | 77.00 | +2.14 | 69.53 | 71.08 | +1.55 |
| RKD [22] | 75.48 | 76.49 | +1.01 | 73.86 | 75.11 | +1.25 | 74.12 | 76.62 | +2.50 | 68.54 | 70.91 | +2.36 |
| PKT [39] | 75.71 | 76.57 | +0.86 | 74.43 | 75.49 | +1.06 | 74.70 | 76.57 | +1.87 | 69.29 | 70.75 | +1.45 |
| AB [40] | 70.12 | 70.76 | +0.64 | 74.38 | 75.51 | +1.13 | 74.73 | 76.51 | +1.78 | 69.76 | 71.05 | +1.29 |
| FT [21] | 75.6 | 76.51 | +0.91 | 74.49 | 75.11 | +0.62 | 74.89 | 77.02 | +2.13 | 69.70 | 71.11 | +1.40 |
| CRD [23] | 75.91 | 77.23 | +1.32 | 74.93 | 76.09 | +1.16 | 75.54 | 76.95 | +1.41 | 70.34 | 71.34 | +1.00 |
| SSKD [26] | 75.96 | 76.80 | +0.84 | 75.72 | 76.03 | +0.31 | 75.95 | 76.85 | +0.90 | 69.34 | 70.29 | +0.96 |
| OH [19] | 76.00 | 76.39 | +0.39 | 74.79 | 75.62 | +0.83 | 75.04 | 76.65 | +1.61 | 68.10 | 69.69 | +1.59 |
| Best | 76.00 | 77.23 | +1.23 | 75.72 | 76.09 | +0.37 | 75.95 | 77.02 | +1.07 | 70.34 | 71.34 | +1.00 |

Table 2: Comparisons between SFTN and the standard teacher models on CIFAR-100 dataset when the architectures of the teacher-student pairs are heterogeneous. In all the tested algorithms, the student models distilled from the teacher models given by SFTN outperform the ones trained from the standard teacher models. All the reported results are based on the outputs of 3 independent runs.

| Teacher/Student Teacher training | resnet32x4/ShuffleV1 | | | resnet32x4/ShuffleV2 | | | ResNet50/VGG8 | | | WRN40-2/ShuffleV2 | | |
|---|---|---|---|---|---|---|---|---|---|---|---|---|
| | Stan. | SFTN | Δ | Stan. | SFTN | Δ | Stan. | SFTN | Δ | Stan. | SFTN | Δ |
| Teacher Acc. | 79.25 | 80.03 | | 79.25 | 79.58 | | 78.70 | 82.52 | | 76.30 | 78.21 | |
| Student Acc. w/o KD | | 71.95 | | | 73.21 | | | 71.12 | | | 73.21 | |
| KD [1] | 74.26 | 77.93 | +3.67 | 75.25 | 78.07 | +2.82 | 73.82 | 74.92 | +1.10 | 76.68 | 78.06 | +1.38 |
| FitNets [18] | 75.95 | 78.75 | +2.80 | 77.00 | 79.68 | +2.68 | 73.22 | 74.80 | +1.58 | 77.31 | 79.21 | +1.90 |
| AT [20] | 76.12 | 78.63 | +2.51 | 76.57 | 78.79 | +2.22 | 73.56 | 74.05 | +0.49 | 77.41 | 78.29 | +0.88 |
| SP [37] | 75.80 | 78.36 | +2.56 | 76.11 | 78.38 | +2.27 | 74.02 | 75.37 | +1.35 | 76.93 | 78.12 | +1.19 |
| VID [38] | 75.16 | 78.03 | +2.87 | 75.70 | 78.49 | +2.79 | 73.59 | 74.76 | +1.17 | 77.27 | 78.78 | +1.51 |
| RKD [22] | 74.84 | 77.72 | +2.88 | 75.48 | 77.77 | +2.29 | 73.54 | 74.70 | +1.16 | 76.69 | 78.11 | +1.42 |
| PKT [39] | 75.05 | 77.46 | +2.41 | 75.79 | 78.28 | +2.49 | 73.79 | 75.17 | +1.38 | 76.86 | 78.28 | +1.42 |
| AB [40] | 75.95 | 78.53 | +2.58 | 76.25 | 78.68 | +2.43 | 73.72 | 74.77 | +1.05 | 77.28 | 78.77 | +1.49 |
| FT [21] | 75.58 | 77.84 | +2.26 | 76.42 | 78.37 | +1.95 | 73.34 | 74.77 | +1.43 | 76.80 | 77.65 | +0.85 |
| CRD [23] | 75.60 | 78.20 | +2.60 | 76.35 | 78.43 | +2.08 | 74.52 | 75.41 | +0.89 | 77.52 | 78.81 | +1.29 |
| SSKD [26] | 78.05 | 79.10 | +1.05 | 78.66 | 79.65 | +0.99 | 76.03 | 76.95 | +0.92 | 77.81 | 78.34 | +0.53 |
| OH [19] | 77.51 | 79.56 | +2.05 | 78.08 | 79.98 | +1.90 | 74.55 | 75.95 | +1.40 | 77.82 | 79.14 | +1.32 |
| Best | 78.05 | 79.56 | +1.51 | 78.66 | 79.98 | +1.32 | 76.03 | 76.95 | +0.92 | 77.82 | 79.21 | +1.39 |

## 4.2 Main Results

To show effectiveness of SFTN, we incorporate SFTN into various existing knowledge distillation algorithms and evaluate accuracy. We present implementation details and experimental results on CIFAR-100 [31] and ImageNet [30] datasets.

### 4.2.1 CIFAR-100

CIFAR-100 [31] consists of 50K training images and 10K testing images in 100 classes. We select 12 state-of-the art distillation methods to compare accuracy of SFTNs with the standard teacher networks. To show the generality of the proposed approach, 8 pairs of teacher and student models have been tested in our experiment. The experiment setup for CIFAR-100 is identical to the one performed in CRD[2]; most experiments employ the SGD optimizer with learning rate 0.05, weight decay 0.0005 and momentum 0.9 while learning rate is set to 0.01 in the ShuffleNet experiments. The hyperparameters for the loss function are set as $\lambda_T = 1$, $\lambda_R^{CE} = 1$, $\lambda_R^{KL} = 3$, and $\tilde{\tau} = 1$ in student-aware training while $\tau = 4$ in knowledge distillation.

Table 1 and 2 demonstrate the full results on the CIFAR-100 dataset. Table 1 presents the distillation performance of all the compared algorithms when teacher and student pairs have the same architecture type while Table 2 shows the results from teacher-student pairs with heterogeneous architecture styles. Both tables clearly demonstrate that SFTN is consistently better than the standard teacher network

---
[2]https://github.com/HobbitLong/RepDistiller

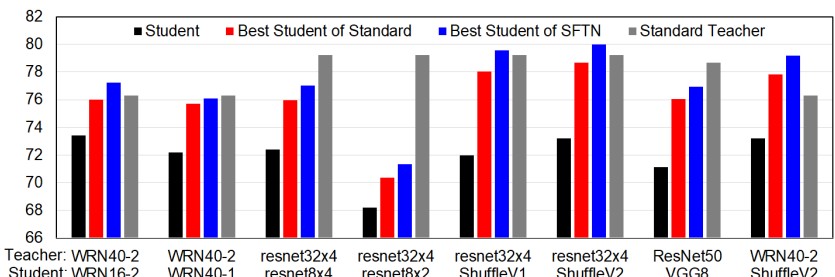

Figure 3: Accuracy comparison of the best students from SFTN with the standard teacher on CIFAR-100. The four best student models of SFTN (blue) outperform the standard teachers (gray) while the only one best student of the standard teacher (red) achieves higher accuracy than its teacher (gray).

Table 3: Top-1 and Top-5 validation accuracy on ImageNet of SFTN in comparison to other methods.

| Teacher/Student | ResNet50/ResNet34 | | | | | |
|---|---|---|---|---|---|---|
| | Top-1 | | | Top-5 | | |
| Teacher training | Standard | SFTN | Δ | Standard | SFTN | Δ |
| Teacher Acc. | 76.45 | 77.43 | | 93.15 | 93.75 | |
| Student Acc. w/o KD | | 73.79 | | | 91.74 | |
| KD [1] | 73.55 | 74.14 | +0.59 | 91.81 | 92.21 | +0.40 |
| FitNets [18] | 74.56 | 75.01 | +0.45 | 92.31 | 92.51 | +0.20 |
| SP [37] | 74.95 | 75.53 | +0.58 | 92.54 | 92.69 | +0.15 |
| CRD [23] | 75.01 | 75.39 | +0.38 | 92.56 | 92.67 | +0.11 |
| OH [19] | 74.56 | 75.01 | +0.45 | 92.36 | 92.56 | +0.20 |
| Best | 75.01 | 75.53 | +0.52 | 92.56 | 92.69 | +0.13 |

in all algorithms. The average difference between SFTN and the standard teacher is 1.58% points, and the average difference between best student accuracy of SFTN and the standard teacher is 1.10% points. We note that the outstanding performance of SFTN is not only driven by the higher accuracy of teacher models achieved by our student-aware learning technique. As observed in Table 1 and 2, the proposed approach often presents substantial improvement compared to the standard distillation methods despite similar or lower teacher accuracies. Refer to Section 4.4 for the further discussion about the relation of accuracy between teacher and student networks.

Figure 3 illustrates the accuracies of the best student models of the standard teacher and SFTN given teacher and student architecture pairs. Despite the small capacity of the students, the best student models of SFTN sometimes outperform the standard teachers while the only one best student of the standard teacher shows higher accuracy than its teacher.

### 4.2.2 ImageNet

ImageNet [30] consists of 1.2M training images and 50K validation images for 1K classes. We adopt the standard Pytorch set-up for ImageNet training for this experiment[3]. The optimization is given by SGD with learning rate 0.1, weight decay 0.0001 and momentum 0.9.

The coefficients of individual loss terms are set as $\lambda_T = 1$, $\lambda_R^{CE} = 1$, and $\lambda_R^{KL} = 1$, where $\tilde{\tau} = 1$. We conduct the ImageNet experiment for 5 different knowledge distillation methods, where teacher models based on ResNet50 transfer knowledge to student networks with ResNet34.

As presented in Table 3, SFTN consistently outperforms the standard teacher network in all settings. The best student accuracy of SFTN achieves the higher top-1 accuracy than the standard teacher model by approximately 0.5% points. This results implies that the proposed algorithm has great potential on large datasets as well.

### 4.3 Comparison with Collaborative and Curriculum Learning Methods

Contrary to traditional knowledge distillation methods based on static pretrained teachers, collaborative learning approaches employ dynamic teacher networks trained jointly with students and curriculum learning methods keep track of the optimization history of teachers for distillation. Table 4 shows that SFTN outperforms the collaborative learning techniques such as DML [4] and KDCL [5];

---

[3]https://github.com/pytorch/examples/tree/master/imagenet

Table 4: Comparision with collaborative and curriculum learning approaches on CIFAR-100. We employ KDCL-Naïve for ensemble logits of KDCL. RCO is based on the one-stage EEI (equal epoch interval) while RCO-EEI-4 adopts 4 anchor points selection with the EEI strategy. Note that both RCO-EEI-4 and SFTN-4 are trained for $240 \times 4$ epochs.

| Teacher | WRN40-2 | | WRN40-2 | | resnet32x4 | | resnet32x4 | | resnet32x4 | | ResNet50 | |
| Student | WRN16-2 | | WRN40-1 | | resnet8x4 | | ShuffleV1 | | ShuffleV2 | | VGG8 | |
| Standard teacher Acc. | 76.30 | | 76.30 | | 79.25 | | 79.25 | | 79.25 | | 78.70 | |
| Student Acc. w/o KD | 73.41 | | 72.16 | | 72.38 | | 71.95 | | 73.21 | | 71.12 | |
| | Stu. | Δ | Stu. | Δ | Stu. | Δ | Stu. | Δ | Stu. | Δ | Stu. | Δ |
| Standard | 75.46 | +2.05 | 73.73 | +1.57 | 73.39 | +1.01 | 74.26 | +2.31 | 75.25 | +2.04 | 73.82 | +2.70 |
| DML [4] | 75.30 | +1.89 | 74.08 | +1.92 | 74.34 | +1.96 | 73.37 | +1.42 | 73.80 | + 0.59 | 73.01 | +1.89 |
| KDCL [5] | 75.45 | +2.04 | 74.65 | +2.49 | 75.21 | +2.83 | 73.98 | + 2.03 | 74.30 | +1.09 | 73.48 | +2.36 |
| RCO [28] | 75.36 | +1.95 | 74.29 | +2.13 | 74.06 | +1.68 | 76.62 | +4.67 | 77.40 | +4.19 | 74.30 | +3.18 |
| SFTN | 76.25 | +2.84 | 75.09 | +2.93 | 76.09 | +3.71 | 77.93 | +5.98 | 78.07 | +4.86 | 74.92 | +3.80 |
| RCO-EEI-4 [28] | 75.69 | +2.28 | 74.87 | +2.71 | 73.73 | +1.35 | 76.97 | +5.02 | 76.89 | +3.68 | 74.24 | +3.12 |
| SFTN-4 | 76.96 | +3.55 | 76.31 | +4.15 | 76.67 | +4.29 | 79.11 | +7.16 | 78.95 | +5.74 | 75.52 | +4.40 |

Table 5: Effect of varying $\tilde{\tau}$ in the KL-divergence loss of the student-aware training tested on CIFAR-100, where $\tau$ for the knowledge distillation is set to 4. The student accuracy is fairly stable over a wide range of the hyperparameter. Note that the accuracies of SFTNs and the student model are rather inversely correlated, which implies that the maximization of teacher models is not necessarily ideal for knowledge distillation.

| | Accuracy of SFTN | | | | | Student accuracy by KD | | | | |
| Teacher | resnet32x4 | | WRN40-2 | | Avg. | resnet32x4 | | WRN40-2 | | Avg. |
| Student | ShuffleV1 | ShuffleV2 | WRN16-2 | WRN40-1 | | ShuffleV1 | ShuffleV2 | WRN16-2 | WRN40-1 | |
| $\tilde{\tau} = 1$ | 81.19 | 80.26 | 78.23 | 78.14 | 78.85 | 76.05 | 77.18 | 76.30 | 74.75 | **75.58** |
| $\tilde{\tau} = 5$ | 81.23 | 81.56 | 79.22 | 78.31 | 79.54 | 75.36 | 75.59 | 76.31 | 73.64 | 75.10 |
| $\tilde{\tau} = 10$ | 81.27 | 81.98 | 78.81 | 78.38 | 79.58 | 74.47 | 75.93 | 75.85 | 73.62 | 74.76 |
| $\tilde{\tau} = 15$ | 81.89 | 81.74 | 79.27 | 78.63 | **79.74** | 74.78 | 75.65 | 75.79 | 73.49 | 74.81 |
| $\tilde{\tau} = 20$ | 81.60 | 81.70 | 78.84 | 78.45 | 79.59 | 74.62 | 75.88 | 75.82 | 74.03 | 74.95 |

the heterogeneous architectures turn out to be effective for mutual learning. On the other hand, the accuracy of SFTN is consistently higher than that of the curriculum learning method, RCO [28], under the same and even harsher training condition in terms of the number of epochs. Although the identification of the optimal checkpoints may be challenging in the trajectory-based learning, SFTN improves its accuracy substantially with more iterations as shown in the results for SFTN-4.

## 4.4 Effect of Hyperparameters

SFTN computes the KL-divergence loss, $\mathcal{L}_R^{KL}$, to minimize the difference between the softmax outputs of teacher and student branches, which involves two hyperparameters, temperature of the softmax function, $\tau$, and weight for KL-divergence loss term, $\lambda_R^{KL}$. We discuss the impact and trade-off issue of the two hyperparameters. In particular, we present our observations that the student-aware learning is indeed helpful to improve the accuracy of student models while maximizing performance of teacher models may be suboptimal for knowledge distillation.

**Temperature of softmax function** The temperature parameter of the KL-divergence loss in (5), denoted by $\tilde{\tau}$, controls the softness of $\tilde{\mathbf{q}}_T$ and $\tilde{\mathbf{q}}_R^i$; as $\tilde{\tau}$ gets higher, the output of the softmax function becomes smoother. Despite the fluctuation in teacher accuracy, student models given by knowledge distillation via SFTN maintain fairly consistent results. Table 5 also shows that the performance of SFTNs and the student models is rather inversely correlated. This result implies that a loosely optimized teacher model may be more effective for knowledge distillation according to this ablation study.

**Weight for KL-divergence loss** The hyperparameter $\lambda_R^{KL}$ facilitates knowledge distillation by making $\tilde{\mathbf{q}}_T$ similar to $\tilde{\mathbf{q}}_R^i$. However, it affects the accuracy of teacher network negatively. Table 6 shows that the average accuracy gaps between SFTNs and the corresponding student models drop gradually as $\lambda_R^{KL}$ increases. One interesting observation is the student accuracy via SFTN with $\lambda_R^{KL} = 10$ in comparison to its counterpart via the standard teacher; even though the standard teacher network is more accurate than SFTN by a large margin, its corresponding student accuracy is lower than that of SFTN.

Table 6: Effect of varying $\lambda_R^{KL}$ in the knowledge distillation via SFTN tested on CIFAR-100. The accuracies of SFTNs and the corresponding students are not correlated while the accuracy gaps of the two models drop as $\lambda_R^{KL}$ increases.

| Teacher | Accuracy of SFTN | | | | | Student accuracy by KD | | | | |
| | resnet32x4 | | WRN40-2 | | Avg. | resnet32x4 | | WRN40-2 | | Avg. |
| Student | ShuffleV1 | ShuffleV2 | WRN16-2 | WRN40-1 | | ShuffleV1 | ShuffleV2 | WRN16-2 | WRN40-1 | |
|---|---|---|---|---|---|---|---|---|---|---|
| $\lambda_R^{KL}=1$ | 81.19 | 80.26 | 78.23 | 78.14 | **79.46** | 76.05 | 77.18 | 76.30 | 74.75 | 76.07 |
| $\lambda_R^{KL}=3$ | 78.70 | 79.80 | 77.83 | 77.57 | 78.48 | 77.36 | 78.56 | 76.20 | 74.71 | **76.71** |
| $\lambda_R^{KL}=6$ | 78.29 | 78.29 | 77.28 | 76.05 | 77.48 | 77.33 | 77.70 | 76.02 | 74.67 | 76.43 |
| $\lambda_R^{KL}=10$ | 73.02 | 75.01 | 75.03 | 73.51 | 74.14 | 75.57 | 76.62 | 74.19 | 73.08 | 74.87 |
| Standard | 79.25 | 79.25 | 76.30 | 76.30 | 77.78 | 74.31 | 75.25 | 75.28 | 73.56 | 74.60 |

Table 7: Effectiveness of knowledge distillation via SFTN when student models have different capacity compared to the one used in the student-aware training. This experiment is conducted on CIFAR-100. The numbers in bold and red denote the best and the second-best results. SB denotes student branch. (M1: resnet8x2, M2: resnet8x4, M3: resnet32x4, M4: WRN16-1, M5:WRN16-2, M6: WRN40-2, M7: ShuffleV2)

| Teacher/Student | WRN40-2/WRN16-2 (M6/M5) | | | | | | | resnet32x4/ResNet8x4 (M3/M2) | | | | | | |
| SB capacity | N/A | Smaller | | Equal/similar | | Larger | | N/A | Smaller | | Equal/similar | | Larger | |
| SB model | N/A | M1 | M4 | M5 | M7 | M3 | M6 | N/A | M4 | M1 | M2 | M7 | M3 | M6 |
|---|---|---|---|---|---|---|---|---|---|---|---|---|---|---|
| SB Acc. | – | 68.19 | 67.10 | 73.41 | 73.21 | 79.25 | 76.30 | – | 67.10 | 68.19 | 72.38 | 73.21 | 79.25 | 76.30 |
| Teacher Acc. | 76.30 | 76.22 | 75.98 | 78.20 | 78.21 | 78.82 | 78.69 | 79.25 | 76.53 | 77.89 | 79.41 | 79.58 | 80.85 | 80.30 |
| KD [1] | 75.46 | 74.67 | 74.73 | **76.25** | 75.68 | 75.56 | 75.63 | 73.39 | 74.71 | 75.19 | **76.09** | 75.82 | 75.19 | 75.03 |
| SP [37] | 75.43 | 74.75 | 75.29 | **76.77** | 76.56 | 76.13 | 76.08 | 74.06 | 75.31 | 75.76 | **76.37** | 76.09 | 75.62 | 75.36 |
| FT [21] | 75.60 | 74.61 | 75.23 | 76.51 | **76.66** | 76.47 | 76.23 | 74.89 | 75.79 | 76.54 | **77.02** | 76.62 | 76.48 | 76.63 |
| CRD [23] | 75.91 | 76.14 | 76.07 | 77.23 | **77.45** | 77.06 | 76.70 | 75.54 | 76.38 | 76.72 | **76.95** | 76.64 | 76.54 | 76.46 |
| SSKD [26] | 75.96 | 74.35 | 74.41 | **76.80** | 76.49 | 76.73 | 76.72 | 75.95 | 75.06 | 75.77 | **76.85** | 76.22 | 76.67 | 76.24 |
| OH [19] | 76.00 | 74.95 | 74,.97 | 76.39 | **76.49** | 76.27 | 76.15 | 75.04 | 75.65 | 75.69 | **76.65** | 76.48 | 76.38 | 76.44 |
| Average | 75.73 | 74.91 | 75.12 | **76.66** | 76.56 | 76.37 | 76.25 | 74.81 | 75.48 | 75.95 | **76.66** | 76.31 | 76.15 | 76.03 |
| Best | 76.00 | 76.14 | 76.07 | 77.23 | **77.45** | 77.06 | 76.72 | 75.95 | 76.38 | 76.72 | **77.02** | 76.64 | 76.67 | 76.63 |

## 4.5 Versatility of SFTN

Although our teacher network obtained from the student-aware training procedure is specialized for a specific student model, it is also effective to transfer knowledge to the students models with substantially different architectures. Table 7 shows that the benefit of our method is also preserved well as long as the student branch has similar capacity to the student models, where the model capacity is defined by the achievable accuracy via independent training without distillation. In addition, it presents that larger students branches are often effective to enhance distillation performance while smaller student branches are not always helpful. In summary, these results imply that a teacher network in SFTN trained for a specific architecture of student network has the potential to transfer its knowledge to other types of student networks.

## 4.6 Use of Pretrained Teachers

The main goal of knowledge distillation is to maximize the benefit in student networks, and the additional training cost may not be critical in many real applications. However, the increase of training cost originated from the student branch of the teacher network is still undesirable. We can sidestep this limitation by adopting pretrained teacher networks in the student-aware training stage. The training cost of SFTN teacher networks is reduced significantly by using pretrained models, and Table 8 presents the tendency clearly. Compared to 240 epochs for the standard student-aware training, fine-tuning pretrained teacher networks only needs 60 epochs for convergence; we train the student branches only for the first 30 epochs and fine-tune the whole network for the remaining 30 epochs. Table 9 shows that fine-tuned pretrained teacher networks have potential to enhance distillation performance. They achieve almost same accuracy with the full SFTN in 6 knowledge distillation algorithms.

## 4.7 Similarity between Teacher and Student Representations

The similarity between teacher and student models is an important measure for knowledge distillation performance in the sense that a student network aims to resemble the output representations of

Table 8: Training time of SFTN teachers with pretrained models. The additional training time in each SFTN with a pretrained teacher model is presented in the last column; it is significantly reduced compared to the standard SFTN teacher (the second-last column).

| Models (teacher/student) | Training time (sec) | | | |
| --- | --- | --- | --- | --- |
| | Teacher | Student | SFTN teacher | SFTN teacher with a pretrained model |
| resnet32x4/ShuffleV1 | 6,005 | 5,624 | 10,910 | 2,298 |
| resnet32x4/ShuffleV2 | 6,005 | 6,221 | 10,949 | 2,370 |
| WRN40-2/WRN16-2 | 3,940 | 1,745 | 6,028 | 1,178 |
| WRN40-2/WRN40-1 | 3,940 | 3,698 | 7,431 | 1,621 |

Table 9: Performance of SFTN fine-tuned from a pretrained teacher (SFTN-FT) on CIFAR-100.

| Teacher/Student | resnet32x4/ShuffleV2 | | |
| --- | --- | --- | --- |
| Teacher training method | Standard | SFTN | SFTN+FT |
| Teacher Acc. | 79.25 | 80.03 | 80.41 |
| Student Acc. w/o KD | | 71.95 | |
| KD [1] | 75.25 | 78.07 | 78.12 |
| SP [37] | 76.11 | 78.38 | 78.51 |
| FT [21] | 76.42 | 78.37 | 77.90 |
| CRD [23] | 76.35 | 78.43 | 78.88 |
| SSKD [26] | 78.66 | 79.65 | 79.15 |
| OH [19] | 78.08 | 79.98 | 79.68 |
| Average | 76.81 | 78.81 | 78.71 |
| Best | 78.66 | 79.98 | 79.68 |

Table 10: Similarity measurements between teachers and students on CIFAR-100.

| Models (teacher/student) | resnet32x4/ShuffleV2 | | | |
| --- | --- | --- | --- | --- |
| Similarity metric | KL-divergence | | CKA | |
| Teacher training method | Standard | SFTN | Standard | SFTN |
| KD [1] | 1.10 | 0.47 | 0.88 | 0.95 |
| FitNets [18] | 0.79 | 0.38 | 0.89 | 0.95 |
| SP [37] | 0.95 | 0.45 | 0.89 | 0.95 |
| VID [38] | 0.88 | 0.45 | 0.88 | 0.95 |
| CRD [23] | 0.81 | 0.43 | 0.88 | 0.95 |
| SSKD [26] | 0.54 | 0.26 | 0.92 | 0.97 |
| OH [19] | 0.85 | 0.37 | 0.90 | 0.96 |
| AVG | 0.84 | 0.39 | 0.89 | 0.96 |

a teacher network. We employ KL-divergence and CKA [41] as similarity metrics, where lower KL-divergence and higher CKA indicate higher similarity.

Table 10 presents the similarities between the representations of a teacher and a student based on ResNet32×4 and ShuffleV2, respectively, which are given by various algorithms on the CIFAR-100 test set. The results show that the distillations from SFTNs always give higher similarity to the student models with respect to the corresponding teacher networks; SFTN reduces the KL-divergence by more than 50% in average while improving the average CKA by 7% points compared to the standard teacher network. The improved similarity of SFTN is natural since it is trained with student branches to obtain student-friendly representations via the KL-divergence loss.

## 5  Conclusion

We proposed a simple but effective knowledge distillation approach by introducing the novel student-friendly teacher network (SFTN). Our strategy sheds a light in a new direction to knowledge distillation by focusing on the stage to train teacher networks. We train teacher networks along with their student branches, and then perform distillation from teachers to students. The proposed strategy turns out to achieve outstanding performance, and can be incorporated into various knowledge distillation algorithms easily. For the demonstration of the effectiveness of our strategy, we conducted comprehensive experiments in diverse environments, which show consistent performance gains compared to the standard teacher networks regardless of architectural and algorithmic variations.

The proposed approach is effective for achieving higher accuracy with reduced model sizes, but it is not sufficiently verified in the unexpected situations with out-of-distribution inputs, domain shifts, lack of training examples, etc. Also, it still rely on a large number of training examples and still has the limitation of high computational cost and potential privacy issue.

## 6  Acknowledgments and Disclosure of Funding

This work was partly supported by Samsung Electronics Co., Ltd. (IO200626-07474-01) and Institute for Information & Communications Technology Promotion (IITP) grant funded by the Korea government (MSIT) [2017-0-01779; XAI, 2021-0-01343; Artificial Intelligence Graduate School Program (Seoul National University)].

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
