# Learning Student-Friendly Teacher Networks for Knowledge Distillation
## *Supplementary Document*

Dae Young Park[*,1], Moon-Hyun Cha[1], Changwook Jeong[1], Dae Sin Kim[1], and Bohyung Han[*,22]

[1]DIT Center, Samsung Electronics, Korea
[2]ECE & ASRI, Seoul National University, Korea
{p30.daeyoung, moonhyun.cha, chris.jeong, daesin.kim}@samsung.com
bhhan@snu.ac.kr

## A  More Analysis

Table 11: Accuracy on CIFAR-100-C [1]. The results are the average of 19 distortion results of the CIFAR-100-C. The SFTN student by KD consistently achieves higher average accuracy than the student by KD.

| Teacher Student | Accuracy of Studnet by KD | | | | | Accuracy of SFTN Student by KD | | | | |
|---|---|---|---|---|---|---|---|---|---|---|
| | resnet32x4 ShuffleV2 | resnet32x4 ShuffleV1 | WRN40-2 WRN16-2 | WRN40-2 WRN40-1 | AVG | resnet32x4 ShuffleV2 | resnet32x4 ShuffleV1 | WRN40-2 WRN16-2 | WRN40-2 WRN40-1 | AVG |
| w/o distortion | 75.30 | 74.10 | 75.69 | 73.69 | 74.70 | 77.52 | 77.83 | 76.13 | 75.11 | **76.65** |
| intensity=1 | 63.84 | 63.84 | 61.83 | 61.29 | 62.70 | 66.01 | 65.15 | 62.35 | 61.60 | **63.78** |
| intensity=2 | 55.49 | 56.16 | 52.49 | 52.88 | 54.26 | 57.68 | 56.20 | 53.30 | 52.63 | **54.95** |
| intensity=3 | 50.34 | 51.08 | 46.91 | 47.50 | 48.96 | 52.26 | 50.82 | 47.79 | 47.31 | **49.55** |
| intensity=4 | 44.02 | 44.35 | 40.45 | 41.10 | 42.48 | 46.71 | 44.31 | 41.28 | 41.03 | **43.33** |
| intensity=5 | 34.29 | 35.28 | 30.66 | 31.66 | 32.97 | 35.37 | 34.08 | 31.39 | 31.74 | **33.15** |

### A.1  Robustness of Data Distribution Shift

Knowledge distillation models are typically deployed on resource-hungry systems that apply to a real-world problem. And out-of-distribution inputs and domain shifts are inevitable problems in knowledge distillation models. So we employed CIFAR-100-C [1] to evaluate the robustness of our models compared to the standard knowledge distillation. Table 11 demonstrate the benefit of SFTN on the CIFAR-100-C dataset, while w/o distortion presents the CIFAR-100 performance of the tested models. The proposed algorithm outperforms the standard knowledge distillation. However, the average margins diminish gradually with an increase in the corruption intensities. This would be partly because highly corrupted data often suffer from the randomness of predictions and the knowledge distillation algorithms are prone to fail in making correct predictions without additional techniques.

### A.2  Transferability

The goal of transfer learning is to obtain versatile representations that adapt well on unseen datasets. To investigate transferability of the student models distilled from SFTN, we perform experiments to transfer the student features learned on CIFAR-100 to STL10 [2] and TinyImageNet [3]. The representations of the examples in CIFAR-100 are obtained from the last student block and frozen during transfer learning, and then we make the features fit to the target datasets using linear classifiers attached to the last student block.

---

[*]Equal contribution

35th Conference on Neural Information Processing Systems (NeurIPS 2021).

Table 12: The accuracy of student models on STL10 [2] and TinyImageNet [3] by transferring knowledge from the models trained on CIFAR-100.

| Models (Teacher/Student) | resnet32x4/ShuffleV2 | | | | | |
|---|---|---|---|---|---|---|
| | CIFAR100 → STL10 | | | CIFAR100 → TinyImageNet | | |
| Teacher training method | Standard | SFTN | Δ | Standard | SFTN | Δ |
| Teacher accuracy | 69.81 | 76.84 | | 31.25 | 40.16 | |
| Student accuracy w/o KD | | 70.18 | | | 33.81 | |
| KD [4] | 67.49 | 73.81 | +6.32 | 30.45 | 37.81 | +7.36 |
| SP [5] | 69.56 | 75.01 | +5.45 | 31.16 | 38.28 | +7.12 |
| CRD [6] | 71.70 | 75.80 | +4.10 | 35.50 | 40.87 | +5.37 |
| SSKD [7] | 74.43 | 77.45 | +3.02 | 38.35 | 42.41 | +4.06 |
| OH [8] | 72.09 | 76.76 | +4.67 | 33.52 | 39.95 | +6.43 |
| AVG | 71.05 | 75.77 | +4.71 | 33.80 | 39.86 | +6.07 |

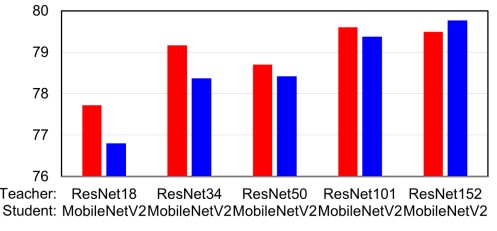

(a) Teacher accuracy on CIFAR-100.

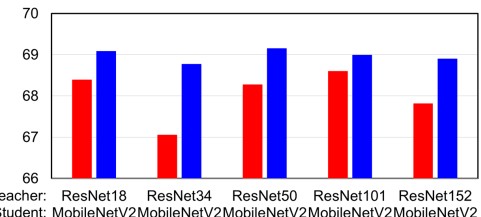

(b) Student Accuracy by KD [4].

Figure 4: Relationship between teacher and student accuracies tested on CIFAR-100, where ResNet with different sizes and MobileNetV2 are employed as teacher and student networks, respectively. In general, the teacher accuracy of SFTN is lower than the standard teacher network, but the student models of SFTN is consistently outperform standard methods.

Table 12 presents transfer learning results on 5 different knowledge distillation algorithms using ResNet32×4 and ShuffleV2 as teacher and student, respectively. Our experiments show that the accuracy of transfer learning on the student models derived from SFTN is consistently better than the students associated with the standard teacher. The average student accuracy of SFTN even outperforms that of the standard teacher by 4.71% points on STL10 [2] and 6.07% points on TinyImageNet [3].

### A.3    Relationship between Teacher and Student Accuracies

Figure 4 demonstrates the relationship between teacher and student accuracies. According to our experiment, higher teacher accuracy does not necessarily lead to better student models. Also, even in the case that the teacher accuracies of SFTN are lower than those of the standard method, the student models of SFTN consistently outperform the counterparts of the standard method. One possible explanation is that SFTN learns adaptive temperatures to the individual elements of a logit. Table 13 shows that teacher networks entropy can be tempered to higher value so that student networks can be more similar to teacher networks by introducing student branch. This result implies that the accuracy gain of a teacher model is not the main reason for the better results of SFTN.

### A.4    Training and Testing Curves

Figure 5(a) illustrates the KL-divergence loss of SFTN for knowledge distillation converges faster than the standard teacher network. This is probably because, by the student-aware training through student branches, SFTN learns better transferrable knowledge to student model than the standard teacher network. We believe that it leads to higher test accuracies of SFTN as shown in Figure 5(b).

### A.5    Additional Study of Hyperparameters

In main paper, we show the effects of $\tau$ and $\lambda_R^{KL}$. However, there are a few more hyperparameters that affect performance in the SFTN framework. So we present additional results of hyperparameters. Table 14 shows that average student accuracy by KD of Branch1+2 is consistently better than the models with a single branch. Also, the average student accuracy by KD of a single branch is higher

Table 13: Shows that the entropy of a teacher given by SFTN is higher than that of a teacher for the standard distillation. This result implies that student-aware training learns adaptive temperatures to the individual elements of a logit, which would be better than the simple temperature scaling by a global constant employed in the standard knowledge distillation.

| Teacher
Student | ResNet18 | ResNet34 | ResNet50
MobilenetV2 | ResNet101 | ResNet152 |
|---|---|---|---|---|---|
| Student training entropy | | | 0.0041 | | |
| Standard teacher training entropy | 0.0004 | 0.0002 | 0.0002 | 0.0002 | 0.0002 |
| SFTN teacher training entropy | 0.0053 | 0.0041 | 0.0044 | 0.0048 | 0.0042 |
| Student accuracy | | | 65.71 | | |
| Standard student accuracy | 68.39 | 67.05 | 68.27 | 68.60 | 68.71 |
| SFTN student accuracy | 69.08 | 68.77 | 69.15 | 68.99 | 68.90 |

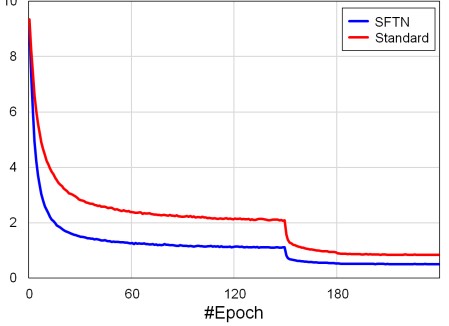

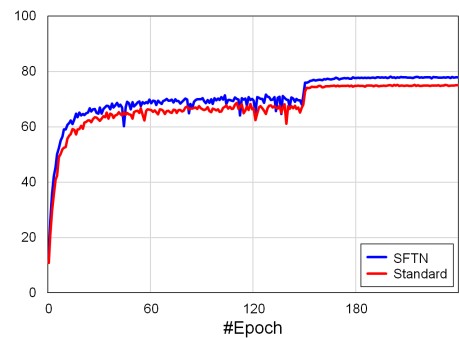

(a) KL-divergence loss during training on CIFAR-100.

(b) Test accuracy on CIFAR-100.

Figure 5: Visualization of training and testing curves on CIFAR-100, where ResNet32×4 and ShuffleV2 are employed as teacher and student networks, respectively. SFTN converges faster and show improved test accuracy than the standard teacher models.

than standard KD. We also test the impacts of $\lambda_R^{CE}$ and $\lambda_T$, which control the weight of cross-entropy loss in the student branch and the teacher network, respectively. Table 15 and 16 show that our results are very consistent for the variations of $\lambda_R^{CE}$ and $\lambda_T$, achieving 76.83±0.10 and 76.55±0.43, respectively, while the accuracy of the standard distillation is 74.60. These additional results with respect to the various hyperparameter settings show the robustness of the SFTN framework.

## A.6 CIFAR-100 Results with Error Bars

To provide variance information from multiple experiments, Table 17 and 18 shows CIFAR-100 results with error bars. The average difference between the error bars for the standard teacher and SFTN is 0.01% points. Therefore, the variance of SFTN is similar to the standard teacher.

## A.7 Additional Study of Similarity

Table 9 of the main paper presents KL-divergence and CKA between the one teacher-student pair (resnet32x4/shuffleV2). To show the generality of the similarity, Table 19 presents additional results of similarity, which illustrate a higher similarity of the teacher given by student-aware training with the corresponding student than the similarity between teacher and student in the standard knowledge distillation. Compared to the standard teacher network, SFTN achieves an average 50% reduction in KL-divergence, a 7% point improvement in average CKA, and an average of 5% higher top 1 agreement.

## B Implementation Details

We present the details of our implementation for better reproduction.

Table 14: Effects of number of student branches on CIFAR-100. Branch1 and Branch2 denote the version that has a single student branch after $F_T^1$ and $F_T^2$, respectively while Branch1+2 indicates the model with student branches after $F_T^1$, $F_T^2$. Refer to Fig. 2(a) of the main paper for the definition of Branch1 and Branch2.

| | Accuracy of SFTN | | | | |
|---|---|---|---|---|---|
| Teacher | resnet32x4 | resnet32x4 | WRN40-2 | WRN40-2 | AVG |
| Student | resnet8x2 | ShuffleV2 | WRN16-2 | ShuffleV2 | |
| Branch1 | 74.61 | 78.06 | 77.19 | 77.34 | 76.80 |
| Branch2 | 75.58 | 76.57 | 75.94 | 75.79 | 75.97 |
| Branch1+2 | 77.89 | 79.58 | 78.20 | 78.21 | **78.47** |
| standard | 79.25 | 79.25 | 76.30 | 76.30 | 77.78 |

(a) Results of teacher network trained with student-aware training

| | Student Accuracy by KD | | | | |
|---|---|---|---|---|---|
| Teacher | resnet32x4 | resnet32x4 | WRN40-2 | WRN40-2 | AVG |
| Student | resnet8x2 | ShuffleV2 | WRN16-2 | ShuffleV2 | |
| Branch1 | 69.46 | 78.11 | 75.66 | 77.23 | 75.12 |
| Branch2 | 69.71 | 76.74 | 75.86 | 77.52 | 74.96 |
| Branch1+2 | 69.17 | 78.07 | 76.25 | 78.06 | **75.39** |
| standard | 67.43 | 75.25 | 75.46 | 76.68 | 73.71 |

(b) Results of student network by KD

Table 15: Effect of $\lambda_R^{CE}$ on CIFAR-100. Student accuracies by KD are consistent for the variation of $\lambda_R^{CE}$.

| | Accuracy of SFTN | | | | |
|---|---|---|---|---|---|
| Teacher | resnet32x4 | resnet32x4 | WRN40-2 | WRN40-2 | AVG |
| Student | resnet8x2 | ShuffleV2 | WRN16-2 | ShuffleV2 | |
| $\lambda_R^{CE} = 1$ | 78.70 | 79.80 | 77.83 | 77.57 | 78.48 |
| $\lambda_R^{CE} = 3$ | 79.71 | 79.98 | 78.41 | 77.94 | **79.01** |
| $\lambda_R^{CE} = 5$ | 79.03 | 79.90 | 77.85 | 78.24 | 78.76 |
| standard | 79.25 | 79.25 | 76.30 | 76.30 | 77.78 |

(a) Results of teacher network trained with student-aware training

| | Student Accuracy by KD | | | | |
|---|---|---|---|---|---|
| Teacher | resnet32x4 | resnet32x4 | WRN40-2 | WRN40-2 | AVG |
| Student | resnet8x2 | ShuffleV2 | WRN16-2 | ShuffleV2 | |
| $\lambda_R^{CE} = 1$ | 77.36 | 78.56 | 76.20 | 74.71 | 76.71 |
| $\lambda_R^{CE} = 3$ | 77.74 | 78.11 | 76.55 | 75.04 | 76.86 |
| $\lambda_R^{CE} = 5$ | 77.73 | 78.05 | 76.45 | 75.40 | **76.91** |
| standard | 74.31 | 75.25 | 75.28 | 73.56 | 74.60 |

(b) Results of student network by KD

## B.1 CIFAR-100

The models for CIFAR-100 are trained for 240 epochs with a batch size of 64, where the learning rate is reduced by a factor of 10 at the 150[th], 180[th], and 210[th] epochs We use randomly cropped 32×32 image with 4-pixel padding and adopt horizontal flipping with a probability of 0.5 for data augmentation. Each channel in an input image is normalized to the standard Gaussian.

## B.2 ImageNet

ImageNet models are learned for 100 epochs with a batch size of 256. We reduce the learning rate by an order of magnitude at the 30[th], 60[th], and 90[th] epochs. In training phase, we perform random cropping with the range from 0.08 to 1.0, which denotes the relative size to the original image while adjusting the aspect ratios by multiplying a random scalar value between 3/4 and 4/3 to the original ratio. All images are resized to 224×224 and flipped horizontally with a probability of 0.5 for data augmentation. In validation phase, images are resized to 256×256, and then center-cropped to 224×224. Each channel in an input image is normalized to the standard Gaussian.

Table 16: Effect of $\lambda_T$ on CIFAR-100. Student accuracies by KD are consistent for the variation of $\lambda_T$.

| | Accuracy of SFTN | | | | |
|---|---|---|---|---|---|
| Teacher | resnet32x4 | resnet32x4 | WRN40-2 | WRN40-2 | AVG |
| Student | resnet8x2 | ShuffleV2 | WRN16-2 | ShuffleV2 | |
| $\lambda_T = 1$ | 78.70 | 79.80 | 77.83 | 77.57 | 78.48 |
| $\lambda_T = 3$ | 80.37 | 81.04 | 78.41 | 78.65 | 79.62 |
| $\lambda_T = 5$ | 80.59 | 81.23 | 78.41 | 78.46 | **79.67** |
| standard | 79.25 | 79.25 | 76.30 | 76.30 | 77.78 |

(a) Results of teacher network trained with student-aware training

| | Student Accuracy by KD | | | | |
|---|---|---|---|---|---|
| Teacher | resnet32x4 | resnet32x4 | WRN40-2 | WRN40-2 | AVG |
| Student | resnet8x2 | ShuffleV2 | WRN16-2 | ShuffleV2 | |
| $\lambda_T = 1$ | 77.36 | 78.56 | 76.20 | 74.71 | 76.71 |
| $\lambda_T = 3$ | 77.66 | 78.33 | 76.68 | 75.22 | **76.97** |
| $\lambda_T = 5$ | 76.45 | 77.14 | 76.19 | 74.75 | 76.13 |
| standard | 74.31 | 75.25 | 75.28 | 73.56 | 74.60 |

(b) Results of student network by KD

Table 17: Comparisons with error bars between SFTN and the standard teacher models on CIFAR-100 dataset when the architectures of the teacher-student pairs are homogeneous. All the reported results are based on the outputs of 3 independent runs.

| Teacher/Student | WRN40-2/WRN16-2 | | WRN40-2/WRN40-1 | | ResNet32x4/ResNet8x4 | | VGG13/VGG8 | |
|---|---|---|---|---|---|---|---|---|
| Teacher training | Standard | SFTN | Standard | SFTN | Standard | SFTN | Standard | SFTN |
| Teacher Acc. | 76.30 | 78.20 | 76.30 | 77.62 | 79.25 | 79.41 | 75.38 | 76.76 |
| Student Acc. w/o KD | 73.41 | | 72.16 | | 72.38 | | 71.12 | |
| KD [4] | 75.46±0.23 | 76.25±0.14 | 73.73±0.21 | 75.09±0.05 | 73.39±0.15 | 76.09±0.32 | 73.41±0.10 | 74.52±0.34 |
| FitNet [9] | 75.72±0.30 | 76.73±0.28 | 74.14±0.58 | 75.54±0.32 | 75.34±0.24 | 76.89±0.09 | 73.49±0.26 | 74.38±0.86 |
| AT [10] | 75.85±0.27 | 76.82±0.24 | 74.56±0.11 | 75.86±0.27 | 74.98±0.12 | 76.91±0.15 | 73.78±0.33 | 73.86±0.15 |
| SP [5] | 75.43±0.24 | 76.77±0.45 | 74.51±0.50 | 75.31±0.48 | 74.06±0.28 | 76.37±0.17 | 73.37±0.16 | 74.62±0.16 |
| VID [11] | 75.63±0.28 | 76.79±0.12 | 74.21±0.05 | 75.76±0.20 | 74.86±0.37 | 77.00±0.22 | 73.81±0.12 | 74.73±0.45 |
| RKD [12] | 75.48±0.45 | 76.49±0.18 | 73.86±0.23 | 75.11±0.14 | 74.12±0.31 | 76.62±0.26 | 73.52±0.20 | 74.48±0.23 |
| PKT [13] | 75.71±0.38 | 76.57±0.22 | 74.43±0.30 | 75.49±0.12 | 74.7±0.33 | 76.57±0.08 | 73.60±0.07 | 74.51±0.13 |
| AB [14] | 70.12±0.18 | 70.76±0.11 | 74.38±0.61 | 75.51±0.07 | 74.73±0.18 | 76.51±0.25 | 73.20±0.26 | 74.67±0.23 |
| FT [15] | 75.6±0.22 | 76.51±0.35 | 74.49±0.41 | 75.11±0.19 | 74.89±0.17 | 77.02±0.15 | 73.64±0.61 | 74.30±0.14 |
| CRD [6] | 75.91±0.25 | 77.23±0.09 | 74.93±0.30 | 76.09±0.47 | 75.54±0.57 | 76.95±0.41 | 74.26±0.37 | 74.86±0.46 |
| SSKD [7] | 75.96±0.03 | 76.80±0.84 | 75.72±0.26 | 76.03±0.15 | 75.95±0.14 | 76.85±0.13 | 74.94±0.24 | 75.66±0.22 |
| OH [8] | 76.00±0.07 | 76.39±0.14 | 74.79±0.19 | 75.62±0.27 | 75.04±0.21 | 76.65±0.11 | 73.94±0.24 | 74.72±0.17 |
| Best | 76.00±0.07 | 77.23±0.09 | 75.72±0.26 | 76.09±0.47 | 75.95±0.14 | 77.02±0.15 | 74.94±0.24 | 75.66±0.22 |

## C  Architecture Details

We present the architectural details of SFTN with VGG13 and VGG8, respectively for teacher and student on CIFAR100. VGG13 and VGG8 are modularized into 4 blocks based on the depths and the feature map sizes. VGG13 SFTN adds a student branch to every output of the teacher network block except the last one. Figure 6, 7 and 8 illustrate the architectures of VGG13 teacher, VGG8 student, and VGG13 SFTN with a VGG8 student branch attached. Table 20, 21 and 22 describe the full details of the architectures.

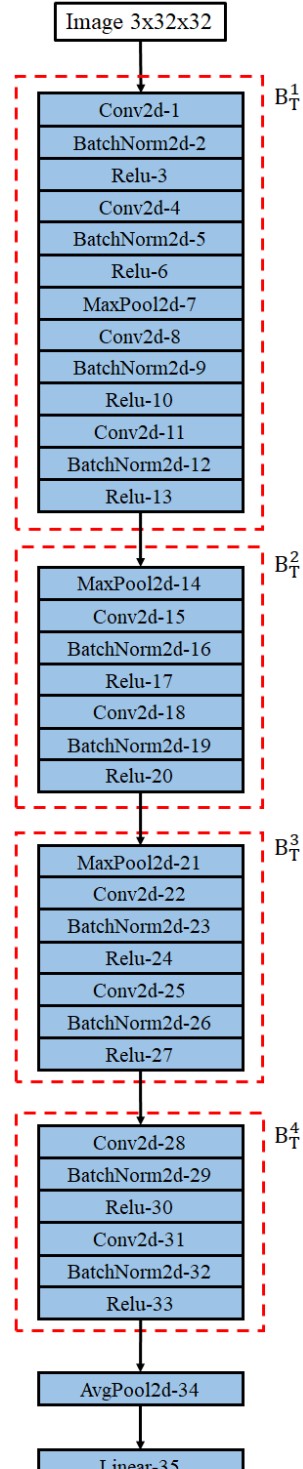

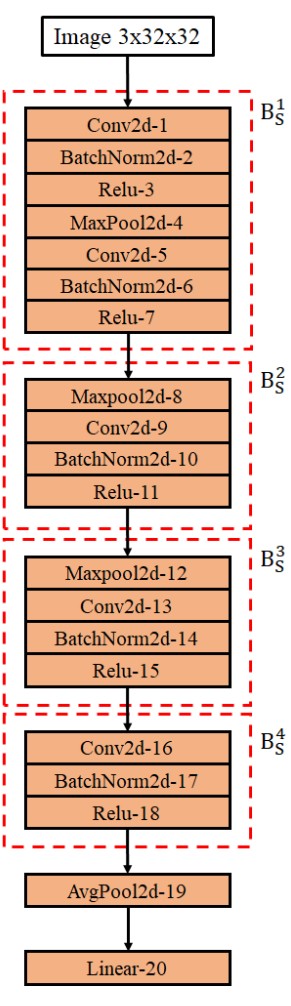

Figure 6: Architecture of VGG13 teacher model. $B_T^i$ and $B_S^i$ denote the $i^{\text{th}}$ block of teacher network and the $i^{\text{th}}$ block of student network, respectively. Table 20 shows detailed description of VGG13 teacher.

Figure 7: Architecture of VGG8 student. $B_T^i$ and $B_S^i$ denote the $i^{\text{th}}$ block of teacher network and the $i^{\text{th}}$ block of student network, respectively. Table 21 shows detailed description of VGG8 student.

Table 18: Comparisons with error bars between SFTN and the standard teacher models on CIFAR-100 dataset when the architectures of the teacher-student pairs are heterogeneous. All the reported results are based on the outputs of 3 independent runs.

| Teacher/Student | ShuffleV1/resnet32x4 | | ShuffleV2/resnet32x4 | | VGG8/ResNet50 | | ShuffleV2/wrn40-2 | |
|---|---|---|---|---|---|---|---|---|
| Teacher training | Standard | SFTN | Standard | SFTN | Standard | SFTN | Standard | SFTN |
| Teacher Acc. | 79.25 | 80.03 | 79.25 | 79.58 | 78.7 | 82.52 | 76.30 | 78.21 |
| Student Acc. w/o KD | 71.95 | | 73.21 | | 71.12 | | 73.21 | |
| KD [4] | 74.26±0.16 | 77.93±0.11 | 75.25±0.05 | 78.07±0.30 | 73.82±0.38 | 74.92±0.35 | 76.68±0.36 | 78.06±0.16 |
| FitNet [9] | 75.95±0.23 | 78.75±0.20 | 77.00±0.19 | 79.68±0.14 | 73.22±0.37 | 74.80±0.21 | 77.31±0.21 | 79.21±0.25 |
| AT [10] | 76.12±0.08 | 78.63±0.27 | 76.57±0.19 | 78.79±0.11 | 73.56±0.25 | 74.05±0.31 | 77.41±0.38 | 78.29±0.14 |
| SP [5] | 75.80±0.29 | 78.36±0.18 | 76.11±0.40 | 78.38±0.38 | 74.02±0.41 | 75.37±0.13 | 76.93±0.07 | 78.12±0.08 |
| VID [11] | 75.16±0.30 | 78.03±0.25 | 75.70±0.40 | 78.49±0.19 | 73.59±0.12 | 74.76±0.37 | 77.27±0.19 | 78.78±0.2 |
| RKD [12] | 74.84±0.23 | 77.72±0.60 | 75.48±0.05 | 77.77±0.39 | 73.54±0.09 | 74.70±0.34 | 76.69±0.23 | 78.11±0.11 |
| PKT [13] | 75.05±0.38 | 77.46±0.14 | 75.79±0.05 | 78.28±0.12 | 73.79±0.06 | 75.17±0.14 | 76.86±0.15 | 78.28±0.13 |
| AB [14] | 75.95±0.20 | 78.53±0.13 | 76.25±0.25 | 78.68±0.22 | 73.72±0.12 | 74.77±0.18 | 77.28±0.24 | 78.77±0.16 |
| FT [15] | 75.58±0.10 | 77.84±0.11 | 76.42±0.45 | 78.37±0.16 | 73.34±0.29 | 74.77±0.42 | 76.80±0.41 | 77.65±0.14 |
| CRD [6] | 75.60±0.09 | 78.20±0.33 | 76.35±0.46 | 78.43±0.06 | 74.52±0.21 | 75.41±0.32 | 77.52±0.39 | 78.81±0.23 |
| SSKD [7] | 78.05±0.15 | 79.10±0.32 | 78.66±0.32 | 79.65±0.05 | 76.03±0.24 | 76.95±0.05 | 77.81±0.19 | 78.34±0.15 |
| OH [8] | 77.51±0.27 | 79.56±0.12 | 78.08±0.18 | 79.98±0.27 | 74.55±0.16 | 75.95±0.12 | 77.82±0.16 | 79.14±0.23 |
| Best | 78.05±0.15 | 79.56±0.12 | 78.66±0.32 | 79.98±0.27 | 76.03±0.24 | 76.95±0.05 | 77.82±0.16 | 79.21±0.25 |

Table 19: Similarity results of various teacher-student pairs. The similarity between teacher and student in the SFTN is consistently higher than standard knowledge distillation.

| Teacher model | resnet32x4 | | resnet32x4 | | WRN40-2 | | WRN40-2 | |
|---|---|---|---|---|---|---|---|---|
| Student model | ShuffleV2 | | ShuffleV1 | | WRN16-2 | | WRN40-1 | |
| Teacher training method | Standard | SFTN | Standard | SFTN | Standard | SFTN | Standard | SFTN |
| KD [4] | 1.10 | 0.47 | 1.08 | 0.43 | 0.72 | 0.26 | 0.91 | 0.35 |
| FitNets [9] | 0.79 | 0.38 | 0.83 | 0.35 | 0.70 | 0.29 | 0.81 | 0.36 |
| SP [5] | 0.95 | 0.45 | 0.90 | 0.37 | 0.07 | 0.26 | 0.80 | 0.32 |
| CRD [6] | 0.81 | 0.43 | 0.85 | 0.40 | 0.65 | 0.26 | 0.77 | 0.31 |
| SSKD [7] | 0.54 | 0.26 | 0.57 | 0.23 | 0.51 | 0.19 | 0.55 | 0.21 |
| OH [8] | 0.85 | 0.37 | 0.78 | 0.30 | 0.69 | 0.23 | 0.75 | 0.27 |
| AVG | 0.84 | 0.39 | 0.84 | 0.35 | 0.66 | 0.25 | 0.77 | 0.30 |

(a) KL divergence results between teacher and student for various combinations of teacher-student architectures and knowledge distillation methods. SFTN consistently generates more similar output distributions than the standard approaches.

| Teacher model | resnet32x4 | | resnet32x4 | | WRN40-2 | | WRN40-2 | |
|---|---|---|---|---|---|---|---|---|
| Student model | ShuffleV2 | | ShuffleV1 | | WRN16-2 | | WRN40-1 | |
| Teacher training method | Standard | SFTN | Standard | SFTN | Standard | SFTN | Standard | SFTN |
| KD [4] | 0.88 | 0.95 | 0.90 | 0.94 | 0.83 | 0.93 | 0.86 | 0.93 |
| FitNets [9] | 0.89 | 0.95 | 0.91 | 0.95 | 0.84 | 0.92 | 0.86 | 0.93 |
| SP [5] | 0.89 | 0.95 | 0.92 | 0.97 | 0.92 | 0.96 | 0.92 | 0.96 |
| CRD [6] | 0.88 | 0.95 | 0.91 | 0.96 | 0.84 | 0.94 | 0.85 | 0.92 |
| SSKD [7] | 0.92 | 0.97 | 0.92 | 0.96 | 0.83 | 0.93 | 0.87 | 0.94 |
| OH [8] | 0.90 | 0.96 | 0.92 | 0.97 | 0.84 | 0.95 | 0.88 | 0.94 |
| AVG | 0.89 | 0.96 | 0.91 | 0.96 | 0.85 | 0.94 | 0.87 | 0.94 |

(b) CKA results between teacher and student for various combinations of teacher-student architectures and knowledge distillation methods. SFTN consistently generates more similar representations than the standard approaches.

| Teacher model | resnet32x4 | | resnet32x4 | | WRN40-2 | | WRN40-2 | |
|---|---|---|---|---|---|---|---|---|
| Student model | ShuffleV2 | | ShuffleV1 | | WRN16-2 | | WRN40-1 | |
| Teacher training method | Standard | SFTN | Standard | SFTN | Standard | SFTN | Standard | SFTN |
| KD [4] | 76.48 | 82.15 | 75.87 | 82.55 | 77.57 | 83.27 | 76.09 | 82.27 |
| FitNets [9] | 78.77 | 83.53 | 77.48 | 83.80 | 77.77 | 82.76 | 76.09 | 82.04 |
| SP [5] | 77.76 | 82.26 | 77.86 | 83.04 | 77.92 | 83.32 | 76.85 | 82.03 |
| CRD [6] | 78.19 | 82.37 | 76.99 | 82.63 | 78.39 | 83.48 | 77.40 | 82.75 |
| SSKD [7] | 82.14 | 85.90 | 82.10 | 86.25 | 79.88 | 85.35 | 79.67 | 85.24 |
| OH [8] | 80.33 | 84.52 | 80.34 | 85.48 | 78.50 | 84.64 | 77.59 | 83.66 |
| AVG | 78.95 | 83.46 | 78.44 | 83.96 | 78.34 | 83.80 | 77.28 | 83.00 |

(c) Top-1 prediction agreement between teacher and student for various combinations of teacher-student architectures and knowledge distillation methods. SFTN consistently achieves higher top-1 agreement than the standard approaches.

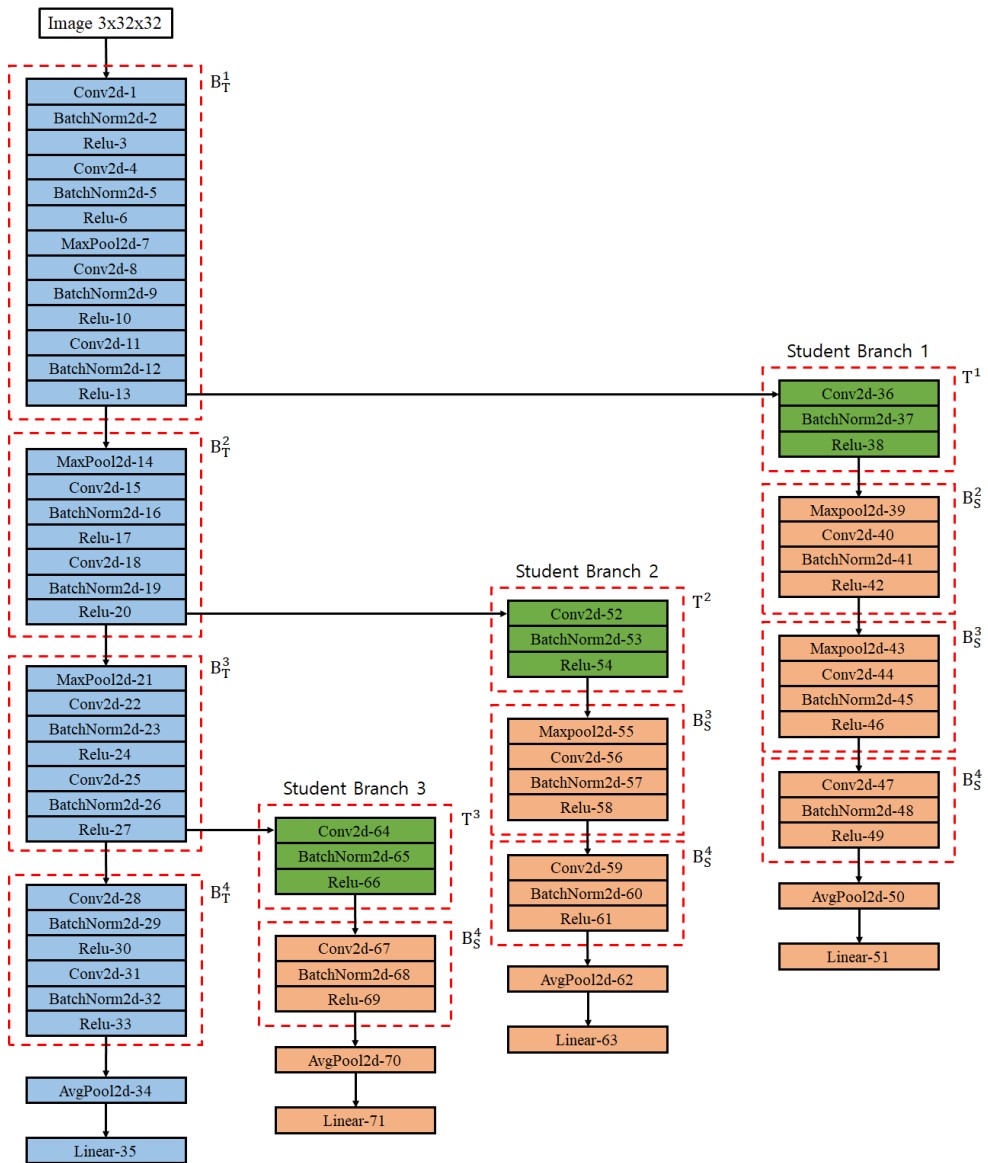

Figure 8: Architecture of SFTN with VGG13 teacher and VGG8 student branch. $B_T^i$, $B_S^i$ and $\mathcal{T}^i$ denote the $i^{\text{th}}$ block of teacher network, the $i^{\text{th}}$ block of student network and teacher network feature transform layer, respectively. Table 22 shows detailed description of VGG13 SFTN attached VGG8 student branch.

Table 20: VGG13 detailed teacher.

| Layer | Input Layer | Input Shape | Filter Size | Channels | Stride | Paddings | Output Shape | Block |
|---|---|---|---|---|---|---|---|---|
| Image | - | - | - | - | - | - | 3x32x32 | - |
| Conv2d-1 | Image | 3x32x32 | 3x3 | 64 | 1 | 1 | 64x32x32 | |
| BatchNorm2d-2 | Conv2d-1 | 64x32x32 | - | 64 | - | - | 64x32x32 | |
| Relu-3 | BatchNorm2d-2 | 64x32x32 | - | - | - | - | 64x32x32 | |
| Conv2d-4 | Relu-3 | 64x32x32 | 3x3 | 64 | 1 | 1 | 64x32x32 | |
| BatchNorm2d-5 | Conv2d-4 | 64x32x32 | - | 64 | - | - | 64x32x32 | |
| Relu-6 | BatchNorm2d-5 | 64x32x32 | - | - | - | - | 64x32x32 | |
| MaxPool2d-7 | Relu-6 | 64x32x32 | 2x2 | - | 2 | 0 | 64x16x16 | $B_T^1$ |
| Conv2d-8 | MaxPool2d-7 | 64x16x16 | 3x3 | 128 | 1 | 1 | 128x16x16 | |
| BatchNorm2d-9 | Conv2d-8 | 128x16x16 | - | 128 | - | - | 128x16x16 | |
| Relu-10 | BatchNorm2d-9 | 128x16x16 | - | - | - | - | 128x16x16 | |
| Conv2d-11 | Relu-10 | 128x16x16 | 3x3 | 128 | 1 | 1 | 128x16x16 | |
| BatchNorm2d-12 | Conv2d-11 | 128x16x16 | - | 128 | - | - | 128x16x16 | |
| Relu-13 | BatchNorm2d-12 | 128x16x16 | - | - | - | - | 128x16x16 | |
| MaxPool2d-14 | Relu-13 | 128x16x16 | 2x2 | - | 2 | 0 | 128x8x8 | |
| Conv2d-15 | MaxPool2d-14 | 128x8x8 | 3x3 | 256 | 1 | 1 | 256x8x8 | |
| BatchNorm2d-16 | Conv2d-15 | 256x8x8 | - | 256 | - | - | 256x8x8 | |
| Relu-17 | BatchNorm2d-16 | 256x8x8 | - | - | - | - | 256x8x8 | $B_T^2$ |
| Conv2d-18 | Relu-17 | 256x8x8 | 3x3 | 256 | 1 | 1 | 256x8x8 | |
| BatchNorm2d-19 | Conv2d-18 | 256x8x8 | - | 256 | - | - | 256x8x8 | |
| Relu-20 | BatchNorm2d-19 | 256x8x8 | - | - | - | - | 256x8x8 | |
| MaxPool2d-21 | Relu-20 | 256x8x8 | 2x2 | - | 2 | 0 | 256x4x4 | |
| Conv2d-22 | MaxPool2d-21 | 256x4x4 | 3x3 | 512 | 1 | 1 | 512x4x4 | |
| BatchNorm2d-23 | Conv2d-22 | 512x4x4 | - | 512 | - | - | 512x4x4 | |
| Relu-24 | BatchNorm2d-23 | 512x4x4 | - | - | - | - | 512x4x4 | $B_T^3$ |
| Conv2d-25 | Relu-24 | 512x4x4 | 3x3 | 512 | 1 | 1 | 512x4x4 | |
| BatchNorm2d-26 | Conv2d-25 | 512x4x4 | - | 512 | - | - | 512x4x4 | |
| Relu-27 | BatchNorm2d-26 | 512x4x4 | - | - | - | - | 512x4x4 | |
| Conv2d-28 | Relu-27 | 512x4x4 | 3x3 | 512 | 1 | 1 | 512x4x4 | |
| BatchNorm2d-29 | Conv2d-28 | 512x4x4 | - | 512 | - | - | 512x4x4 | |
| Relu-30 | BatchNorm2d-29 | 512x4x4 | - | - | - | - | 512x4x4 | $B_T^4$ |
| Conv2d-31 | Relu-30 | 512x4x4 | 3x3 | 512 | 1 | 1 | 512x4x4 | |
| BatchNorm2d-32 | Conv2d-31 | 512x4x4 | - | 512 | - | - | 512x4x4 | |
| Relu-33 | BatchNorm2d-32 | 512x4x4 | - | - | - | - | 512x4x4 | |
| AvgPool2d-34 | Relu-33 | 512x4x4 | - | - | - | - | 512x1x1 | - |
| Linear-35 | AvgPool2d-34 | 512x1x1 | - | - | - | - | 100 | - |

Table 21: VGG8 student model.

| Layer | Input Layer | Input Shape | Filter Size | Channels | Stride | Paddings | Output Shape | Block |
|---|---|---|---|---|---|---|---|---|
| Image | - | - | - | - | - | - | 3x32x32 | - |
| Conv2d-1 | Image | 3x32x32 | 3x3 | 64 | 1 | 1 | 64x32x32 | |
| BatchNorm2d-2 | Conv2d-1 | 64x32x32 | - | 64 | - | - | 64x32x32 | |
| Relu-3 | BatchNorm2d-2 | 64x32x32 | - | - | - | - | 64x32x32 | |
| MaxPool2d-4 | Relu-3 | 64x32x32 | 2x2 | - | 2 | 0 | 64x16x16 | $B_S^1$ |
| Conv2d-5 | MaxPool2d-4 | 64x16x16 | 3x3 | 128 | 1 | 1 | 128x16x16 | |
| BatchNorm2d-6 | Conv2d-5 | 128x16x16 | 2x2 | 128 | 1 | - | 128x16x16 | |
| Relu-7 | BatchNorm2d-6 | 128x16x16 | - | - | - | - | 128x16x16 | |
| Maxpool2d-8 | Relu-7 | 128x16x16 | 2x2 | - | 2 | 0 | 128x8x8 | |
| Conv2d-9 | Maxpool2d-8 | 128x8x8 | 3x3 | 256 | 1 | 1 | 256x8x8 | $B_S^2$ |
| BatchNorm2d-10 | Conv2d-9 | 256x8x8 | - | 256 | - | - | 256x8x8 | |
| Relu-11 | BatchNorm2d-10 | 256x8x8 | - | - | - | - | 256x8x8 | |
| MaxPool2d-12 | Relu-11 | 256x8x8 | 2x2 | - | 2 | 0 | 256x4x4 | |
| Conv2d-13 | MaxPool2d-12 | 256x4x4 | 3x3 | 512 | 1 | 1 | 512x4x4 | $B_S^3$ |
| BatchNorm2d-14 | Conv2d-13 | 512x4x4 | - | 512 | - | - | 512x4x4 | |
| Relu-15 | BatchNorm2d-14 | 512x4x4 | - | - | - | - | 512x4x4 | |
| Conv2d-16 | Relu-15 | 512x4x4 | 3x3 | 512 | 1 | 1 | 512x4x4 | |
| BatchNorm2d-17 | Conv2d-16 | 512x4x4 | - | 512 | - | - | 512x4x4 | $B_S^4$ |
| Relu-18 | BatchNorm2d-17 | 512x4x4 | - | - | - | - | 512x4x4 | |
| AvgPool2d-19 | Relu-18 | 512x4x4 | - | - | - | - | 512x1x1 | - |
| Linear-20 | AvgPool2d-19 | 512x1x1 | - | - | - | - | 100 | - |

Table 22: Details of SFTN architecture with VGG13 teacher and VGG8 student branch.

| Layer | Input Layer | Input Shape | Filter Size | Channels | Stride | Paddings | Output Shape | Block |
|---|---|---|---|---|---|---|---|---|
| Image | - | - | - | - | - | - | 3x32x32 | - |
| Conv2d-1 | Image | 3x32x32 | 3x3 | 64 | 1 | 1 | 64x32x32 | |
| BatchNorm2d-2 | Conv2d-1 | 64x32x32 | - | 64 | - | - | 64x32x32 | |
| Relu-3 | BatchNorm2d-2 | 64x32x32 | - | - | - | - | 64x32x32 | |
| Conv2d-4 | Relu-3 | 64x32x32 | 3x3 | 64 | 1 | 1 | 64x32x32 | |
| BatchNorm2d-5 | Conv2d-4 | 64x32x32 | - | 64 | - | - | 64x32x32 | |
| Relu-6 | BatchNorm2d-5 | 64x32x32 | - | - | - | - | 64x32x32 | |
| MaxPool2d-7 | Relu-6 | 64x32x32 | 2x2 | - | 2 | 0 | 64x16x16 | $B_T^1$ |
| Conv2d-8 | MaxPool2d-7 | 64x16x16 | 3x3 | 128 | 1 | 1 | 128x16x16 | |
| BatchNorm2d-9 | Conv2d-8 | 128x16x16 | - | 128 | - | - | 128x16x16 | |
| Relu-10 | BatchNorm2d-9 | 128x16x16 | - | - | - | - | 128x16x16 | |
| Conv2d-11 | Relu-10 | 128x16x16 | 3x3 | 128 | 1 | 1 | 128x16x16 | |
| BatchNorm2d-12 | Conv2d-11 | 128x16x16 | - | 128 | - | - | 128x16x16 | |
| Relu-13 | BatchNorm2d-12 | 128x16x16 | - | - | - | - | 128x16x16 | |
| MaxPool2d-14 | Relu-13 | 128x16x16 | 2x2 | - | 2 | 0 | 128x8x8 | |
| Conv2d-15 | MaxPool2d-14 | 128x8x8 | 3x3 | 256 | 1 | 1 | 256x8x8 | |
| BatchNorm2d-16 | Conv2d-15 | 256x8x8 | - | 256 | - | - | 256x8x8 | |
| Relu-17 | BatchNorm2d-16 | 256x8x8 | - | - | - | - | 256x8x8 | $B_T^2$ |
| Conv2d-18 | Relu-17 | 256x8x8 | 3x3 | 256 | 1 | 1 | 256x8x8 | |
| BatchNorm2d-19 | Conv2d-18 | 256x8x8 | - | 256 | - | - | 256x8x8 | |
| Relu-20 | BatchNorm2d-19 | 256x8x8 | - | - | - | - | 256x8x8 | |
| MaxPool2d-21 | Relu-20 | 256x8x8 | 2x2 | - | 2 | 0 | 256x4x4 | |
| Conv2d-22 | MaxPool2d-21 | 256x4x4 | 3x3 | 512 | 1 | 1 | 512x4x4 | |
| BatchNorm2d-23 | Conv2d-22 | 512x4x4 | - | 512 | - | - | 512x4x4 | |
| Relu-24 | BatchNorm2d-23 | 512x4x4 | - | - | - | - | 512x4x4 | $B_T^3$ |
| Conv2d-25 | Relu-24 | 512x4x4 | 3x3 | 512 | 1 | 1 | 512x4x4 | |
| BatchNorm2d-26 | Conv2d-25 | 512x4x4 | - | 512 | - | - | 512x4x4 | |
| Relu-27 | BatchNorm2d-26 | 512x4x4 | - | - | - | - | 512x4x4 | |
| Conv2d-28 | Relu-27 | 512x4x4 | 3x3 | 512 | 1 | 1 | 512x4x4 | |
| BatchNorm2d-29 | Conv2d-28 | 512x4x4 | - | 512 | - | - | 512x4x4 | |
| Relu-30 | BatchNorm2d-29 | 512x4x4 | - | - | - | - | 512x4x4 | |
| Conv2d-31 | Relu-30 | 512x4x4 | 3x3 | 512 | 1 | 1 | 512x4x4 | $B_T^4$ |
| BatchNorm2d-32 | Conv2d-31 | 512x4x4 | - | 512 | - | - | 512x4x4 | |
| Relu-33 | BatchNorm2d-32 | 512x4x4 | - | - | - | - | 512x4x4 | |
| AvgPool2d-34 | Relu-33 | 512x4x4 | - | - | - | - | 512x1x1 | - |
| Linear-35 | AvgPool2d-34 | 512x1x1 | - | - | - | - | 100 | - |
| Student Branch 1 | | | | | | | | |
| Conv2d-36 | Relu-13 | 128x16x16 | 1x1 | 128 | 1 | 0 | 128x16x16 | |
| BatchNorm2d-37 | Conv2d-36 | 128x16x16 | - | 128 | - | - | 128x16x16 | $\mathcal{T}^1$ |
| Relu-38 | BatchNorm2d-37 | 128x16x16 | - | - | - | - | 128x16x16 | |
| Maxpool2d-39 | BatchNorm2d-37 | 128x16x16 | 2x2 | - | 2 | 0 | 128x8x8 | |
| Conv2d-40 | Maxpool2d-39 | 128x8x8 | 3x3 | 256 | 1 | 1 | 256x8x8 | $B_S^2$ |
| BatchNorm2d-41 | Conv2d-40 | 256x8x8 | - | 256 | - | - | 256x8x8 | |
| Relu-42 | BatchNorm2d-41 | - | - | - | - | - | 256x8x8 | |
| MaxPool2d-43 | Relu-42 | 256x8x8 | 2x2 | - | 2 | 0 | 256x4x4 | |
| Conv2d-44 | MaxPool2d-43 | 256x4x4 | 3x3 | 512 | 1 | 1 | 512x4x4 | $B_S^3$ |
| BatchNorm2d-45 | Conv2d-44 | 512x4x4 | - | 512 | - | - | 512x4x4 | |
| Relu-46 | BatchNorm2d-45 | - | - | - | - | - | 512x4x4 | |
| Conv2d-47 | Relu-46 | 512x4x4 | 3x3 | 512 | 1 | 1 | 512x4x4 | |
| BatchNorm2d-48 | Conv2d-47 | 512x4x4 | - | 512 | - | - | 512x4x4 | $B_S^4$ |
| Relu-49 | BatchNorm2d-48 | - | - | - | - | - | 512x4x4 | |
| AvgPool2d-50 | Relu-49 | 512x4x4 | - | - | - | - | 512x1x1 | - |
| Linear-51 | AvgPool2d-50 | 512x1x1 | - | - | - | - | 100 | - |

(continued from the previous table)

| Layer | Input Layer | Input Shape | Filter Size | Channels | Stride | Paddings | Output Shape | Block |
|---|---|---|---|---|---|---|---|---|
| | | | Student Branch 2 | | | | | |
| Conv2d-52 | Relu-20 | 256x8x8 | 1x1 | 256 | 1 | 0 | 256x8x8 | |
| BatchNorm2d-53 | Conv2d-52 | 256x8x8 | - | 256 | - | - | 256x8x8 | $\mathcal{T}^2$ |
| Relu-54 | BatchNorm2d-53 | 256x8x8 | - | - | - | - | 256x8x8 | |
| MaxPool2d-55 | Relu-54 | 256x8x8 | 2x2 | - | 2 | 0 | 256x4x4 | |
| Conv2d-56 | MaxPool2d-55 | 256x4x4 | 3x3 | 512 | 1 | 1 | 512x4x4 | $B_S^3$ |
| BatchNorm2d-57 | Conv2d-56 | 512x4x4 | - | 512 | - | - | 512x4x4 | |
| Relu-58 | BatchNorm2d-57 | - | - | - | - | - | 512x4x4 | |
| Conv2d-59 | Relu-58 | 512x4x4 | 3x3 | 512 | 1 | 1 | 512x4x4 | |
| BatchNorm2d-60 | Conv2d-59 | 512x4x4 | - | 512 | - | - | 512x4x4 | $B_S^4$ |
| Relu-61 | BatchNorm2d-60 | - | - | - | - | - | 512x4x4 | |
| AvgPool2d-62 | Relu-61 | 512x4x4 | - | - | - | - | 512x1x1 | - |
| Linear-63 | AvgPool2d-62 | 512x1x1 | - | - | - | - | 100 | - |
| | | | Student Branch 3 | | | | | |
| Conv2d-64 | Relu-27 | 512x4x4 | 1x1 | 512 | 1 | 0 | 512x4x4 | |
| BatchNorm2d-65 | Conv2d-64 | 512x4x4 | - | 512 | - | - | 512x4x4 | $\mathcal{T}^3$ |
| Relu-66 | BatchNorm2d-65 | 512x4x4 | - | - | - | - | 512x4x4 | |
| Conv2d-67 | Relu-66 | 512x4x4 | 3x3 | 512 | 1 | 1 | 512x4x4 | |
| BatchNorm2d-68 | Conv2d-67 | 512x4x4 | - | 512 | - | - | 512x4x4 | $B_S^4$ |
| Relu-69 | BatchNorm2d-68 | - | - | - | - | - | 512x4x4 | |
| AvgPool2d-70 | Relu-69 | 512x4x4 | - | - | - | - | 512x1x1 | - |
| Linear-71 | AvgPool2d-70 | 512x1x1 | - | - | - | - | 100 | - |