# OpenReview forum: "Learning Student-Friendly Teacher Networks for Knowledge Distillation"
_NeurIPS.cc/2021/Conference — NeurIPS 2021 Poster_

### Official Review · Reviewer_9tKz · 2021-07-12

**Rating:** 5
**Confidence:** 4

**Summary:**

This paper proposes a student friendly teacher network (SFTN) strategy for knowledge distillation. In experiments, the proposed method boost performance for various current knowledge distillation algorithms on CIFAR-100 and ImageNet.


**Limitations And Societal Impact:**

See weakness. I am pleased to improve the final rating if the authors can address all my concerns.

**Main Review:**

Strength:
1. The authors propose a student friendly teacher network (SFTN) for knowledge distillation. The proposed method is orthogonal to the current KD methods.
2. The experimental results are strong. SFTN demonstrates the performance improvements for various KD methods on CIFAR-100 and ImageNet.
3. The writing and presentation are clear and well-organized.

Weakness:
1. The propose SFTN is a little inefficient. According to the Sec. 3.1, for each student model, SFTN needs to train a specialized teacher model. Although Sec. 4.6 and Table 8 give some analyses on CIFAR-100, I still think the proposed SFTN may not be flexible and efficient, especially for the large-scale training data (i.e. ImageNet) and the huge teacher models (i.e. ResNet-101/ResNeXt-101). It is necessary to discuss the training time cost for SFTN.
2. According to the experiments, SFTN got significant performance improvements (1~2%) on CIFAR-100, but marginal improvements (0.5%) on ImageNet. Can the authors give some theoretical or empirical explanations for this phenomenon? Or it means that, with the increasing number of training data, SFTN may not be effect to keep the consistency between teacher and student features.
3. The experiments on ImageNet are not sufficient. Considering the network structures and the capacity, ResNet-50 is similar to ResNet-34. The authors should give more experiments on the large gap of teacher-student models for homogeneous architecture (ResNet101->ResNet18) and heterogeneous architecture (ResNet101->VGG-16) to support the authors' claim (L24-26).

**Time Spent Reviewing:**

4

---

### Official Review · Reviewer_Bfa6 · 2021-07-18

**Rating:** 5
**Confidence:** 4

**Summary:**

This paper proposes SFTN to learn teacher models that are better at transferring their knowledge to the student models (i.e., student-friendly). The proposed method makes the teacher model aware of the student models during their training process by jointly learning the student branches. The proposed method is versatile to different kinds of KD algorithms. Experiments on image classification benchmarks show that SFTN successfully improves the performance of different KD methods. The authors also conduct ablation studies to better understand the effectiveness of the proposed method.

**Limitations And Societal Impact:**

Limited discussion about very related work on online KD literature and limited discussion about the computational cost for training the teacher model.

**Main Review:**

## Pros

- The paper is well written and the problem of the student-teacher capacity gap is an important problem and the idea of learning student-friendly teacher models is well motivated.

- Experiments are well conducted and the results seem to be promising. SFTN consistently improves various KD methods in many different settings.

## Cons

- The novelty of the proposed paper is limited because the line of online knowledge distillation research is conceptually very similar to the SFTN method. They learn teacher and student models together and the objective function is very similar to the case of SFTN, except that they do not add the second stage KD because they focus on one-stage KD, and they do not introduce branches of student models. A baseline that trains the student model jointly with the teacher model, but separates the students and teachers (independent students rather than student branches) may be helpful.

- The performance of the teacher model trained by SFTN is higher than a conventional teacher in many cases, therefore it is hard to tell if the SFTN model is student-friendly or simply stronger and generalizes better.

- The computational overhead of training the teacher model by SFTN should be mentioned. Since most KD method assumes a pre-trained teacher model, the fact that SFTN requires re-train the teacher model in a more sophisticated way is obviously a drawback. What if we train a new powerful teacher with the same computational budget compared to that of SFTN teacher training?

-The choice of student branches is non-trivial. Also, how are the student branches possible if the student and teacher have completely different architectures?





**Time Spent Reviewing:**

4

---

### Official Review · Reviewer_6tca · 2021-07-20

**Rating:** 4
**Confidence:** 4

**Summary:**

This work proposes Student-Friendly Teacher Network (SFTN), an approach that aims to learn teacher models that are friendly to student models, thereby facilitating the subsequent knowledge distillation process resulting in superior student models (studied image classification problems).



**Limitations And Societal Impact:**

Yes, limitations are briefly discussed in Sec 5.


**Main Review:**

The work is more of an engineering work supported by large scale empirical study. The ideas and results are not particularly novel nor exciting. Considering this as an engineering work, the practicality/ usability of the proposed technique may be limited due to the large computational overhead (i.e.: using student branches in SFTN training).

Paper Strengths:

The paper has a clear definition of the problem, proposed methodology and extensive empirical studies.

Paper Weaknesses:

Is $L_{R}^{CE}$ term required during SFTN training? Given that the objective of SFTN training is to make the teacher and student representations similar to each other (this is what is meant by friendly), do the authors think that the use of $L_T$ and $L_{R}^{KL}$ is sufficient?

For feature-based KD, how did the authors decide on the design for modularizing the teacher and student networks since there are multiple design choices and this will also vary for every pair of (Teacher, Student) setups.

KD is done on the training set. So for Table 9, the similarities between the representations must be measured using the training set.

Did the authors observe identical behaviour with respect to similarities between representations (Table 9) in ImageNet experiments (I don’t see these results). I emphasize this point as this work focuses on representation learning and ImageNet is more representative of challenging learning problems.

Given that authors propose a KD framework, during distillation (training of student) do authors learn only from the teacher (soft-targets) or from both the ground truth (hard-targets) and the teacher (soft-targets)? I think this is an important detail to clarify as I expect that the authors only use the soft-targets for distillation.

Why should the student branches learn multiple representations ( $q_{R}^{1}, q_{R}^{2} )$? How does the student's accuracy change if only $q_{R}^{1}$ is used?

Giving more context into the term "friendly representations" could be better. I.e.: In these setups can we say that KL divergence between the teacher and student representations can be used to measure this degree of friendliness?


Clarity:

The paper is ok. One suggestion is to move some hyper-parameter details to Supplementary so that it can be less distracting to read (i.e.: learning rates, optimizer details etc).


Relation to prior work:

Yes, the contribution of this work is discussed




**Time Spent Reviewing:**

3 hours

---

### Official Review · Reviewer_rkNh · 2021-07-21

**Rating:** 4
**Confidence:** 4

**Summary:**

This paper proposes a simple method to facilitate knowledge transfer from a teacher to a student. Unlike traditional knowledge distillation methodology that aims to effectively train student networks given pre-trained teachers, the goal of the work is training teachers which are friendly to knowledge transfer to students. The authors propose a framework in which the target student network is incorporated into teacher networks through a set of student branches. The teacher network and associated student branches are trained jointly before distillation. Extensive experiments on CIFAR and ImageNet datasets are conducted for evaluation.

**Limitations And Societal Impact:**

The authors have discussed some limitations of the work in the conclusion section.

**Main Review:**

This paper is well presented and organized. The motivation of the work is interesting. Unlike effectively training students given pretrained teachers, the authors take a different perspective that expects to learn a student-friendly teacher which facilitates knowledge transfer to the target student. Extensive experiments and ablation studies, including comparison to collaborative learning methods, on CIFAR and ImageNet datasets are conducted for evaluation.

However, the proposed solution is less convincing as the target student network architecture needs to be given ahead, which may reduce the generalization of knowledge distillation. Given a pretrained large model, the distillation typically performs without any assumption or restriction on target network architectures and can work on any target networks in practice. It may happen that the proposed method can only work well on given pairs of teachers and students but its benefit may degrade when the student networks for training SFTN and distillation are dramatically different. The authors need to analyze and provide empirical studies for the problem.

The technical novelty of the work is incremental as the network design and loss functions are based on existing methods. The design of adding multiple (typically shallow) branches into deep networks is highly related to deeply supervised learning [1] which introduces extra branches to facilitate the training of deep networks. The relation to the work needs to be discussed although the motivation of the two works are different.

Some questions about the experimental section:
1)	I think resnet32x4 and resnet8x4 denotes the network configuration of ResNext rather than ResNet. Otherwise, the definition of the two networks needs to be clarified in the paper.
2)	The explanation of experimental results are insufficient. For example, lack of analyzing the phenomenon that the teacher performance of SFTN in the setting of WRN40-2/WRN16-2 is much better than WRN40-2/WRN40-1. I think the reason is that using shallow branches may benefit the training of deep networks. The authors need to improve the analysis in the experimental section.

[1] Chen-Yu Lee et al., Deeply-Supervised Nets. In AISTATS 2015.


**Time Spent Reviewing:**

4

---

### Public Comment · ~Greg_Adams1 · 2022-08-22
**Thanks!**

Hello! I need to educate you one regarding the research paper writing service https://www.bestessayservicereviews.com/ that will take care of your concern with composing a paper and some other composed work. Your work will be destined to be of the best quality, without copyright infringement and on time. It will save you time and further develop your school execution.

---

### Decision · Program_Chairs · 2021-09-28

**Decision:**

Accept (Poster)

**Comment:**

Reviewers are in broad agreement, finding the paper below the bar for NeurIPS. Reviewers found the problem itself interesting, and the experiments reasonably thorough. Criticisms centered around issues of novelty, efficiency, and additional components that should be added to the experiments to make them more convincing. None of these individually seemed like deal-breakers, but collectively put the paper somewhat below the threshold.

**Consistency Experiment:**

NeurIPS has a long history of experimentation. In 2014, NeurIPS ran an experiment in which 10% of submissions were reviewed by two independent committees to quantify the randomness in the review process. This year, we repeated a variant of this experiment to see how the quality of the review process has changed over time.  This paper was part of the experiment and was therefore assigned to two committees (consisting of reviewers, an Area Chair, and a Senior Area Chair) that reached independent decisions.  If both committees made the same recommendation, this recommendation was followed. If a single committee recommended acceptance, the paper was accepted (with the exception of a few cases in which the other committee identified what we considered a fatal flaw, e.g., an error in a key result).

This copy’s committee reached the following decision: **Reject**

The other committee assigned to the paper recommended **Accept (Poster)**.  You can find the other set of reviews, along with any follow up discussion with the authors here:
https://openreview.net/forum?id=sqZ-b0a6Wm